

# The CALIPSO version 4.5 stratospheric aerosol subtyping algorithm

Jason L. Tackett[1], Jayanta Kar[2], Mark A. Vaughan[1], Brian J. Getzewich[1], Man-Hae Kim[3], Jean-Paul Vernier[4], Ali H. Omar[1], Brian E. Magill[2], Michael C. Pitts[1], David M. Winker[1]

[1]NASA Langley Research Center, Hampton, VA, USA
[2]Science Systems and Applications, Inc., Hampton, VA, USA
[3]Seoul National University, Seoul, South Korea
[4]National Institute of Aerospace Associates, Hampton, VA, USA

*Correspondence to*: Jason L. Tackett (jason.l.tackett@nasa.gov)

**Abstract.**

The accurate classification of aerosol types injected into the stratosphere is important to properly characterize their chemical and radiative impacts within the Earth climate system. The updated stratospheric aerosol subtyping algorithm used in the Version 4.5 (V4.5) release of the Cloud Aerosol Lidar with Orthogonal Polarization (CALIOP) level 2 data products now delivers more comprehensive and accurate classifications than its predecessor. The original algorithm identified four aerosol subtypes for layers detected above the tropopause: volcanic ash, smoke, sulfate/other, and polar stratospheric aerosol (PSA). In the revised algorithm, sulfates are separately identified as a distinct, homogeneous subtype and the diffuse, weakly scattering layers previously assigned to the sulfate/other class are recategorized as a fifth "unclassified" subtype. By making two structural changes to the algorithm and revising two thresholds, the V4.5 algorithm improves the ability to discriminate between volcanic ash and smoke from pyrocumulonimbus injections, improves the fidelity of the sulfate subtype, and more accurately reflects the uncertainties inherent in the classification process. The 532 nm lidar ratio for volcanic ash was also revised to a value more consistent with the current state of knowledge. This paper briefly reviews the previous version of the algorithm (V4.1/V4.2), then fully details the rationale and impact of the V4.5 changes on subtype classification frequency for specific events where the dominant aerosol type is known based on literature. Classification accuracy is best for volcanic ash due to its characteristically high depolarization ratio. Smoke layers in the stratosphere are also classified with reasonable accuracy, though during the daytime a substantial fraction are misclassified as ash. It is also possible for mixtures of ash and sulfate to be misclassified as smoke. The V4.5 sulfate subtype accuracy is less than that for ash or smoke, with sulfates being misclassified as smoke about one-third of the time. However, because exceptionally tenuous layers are now assigned to the unclassified subtype and the revised algorithm levies more stringent criteria for identifying an aerosol as sulfate, it is more likely that layers labeled as this subtype are in fact sulfate.



## 1 Introduction

Injections of aerosol into the stratosphere have important impacts on the chemistry in the upper atmosphere and affect the Earth's radiative energy balance (Kremser et al., 2016). Explosive volcanic eruptions can inject large amounts of ash and sulfur dioxide ($SO_2$) gas into the lower and middle stratosphere. Ash, which is most often composed of silicates, can remain in the atmosphere for weeks or, in extreme cases months, as in the case of the 2014 Kelud eruption (Vernier et al., 2016). $SO_2$ reacts with the hydroxyl radical OH in the stratosphere through photochemistry, forming sulfuric acid ($H_2SO_4$) nuclei

which grow by condensation and coagulation into larger sulfate aerosols (Kremser et al., 2016). The radiative and chemical impacts of sulfate in the stratosphere can be significant (e.g., Stone et al., 2017). A third aerosol type in the upper troposphere and lower stratosphere (UTLS) is becoming widely recognized as important: smoke from pyrocumulonimbus (pyroCb) events within intense wildfires (Fromm et al., 2010). PyroCb events inject smoke to altitudes of 20 km or higher due to the buoyancy from the intense heat of the fire and meteorological conditions that favor the development of deep

convection, specifically moisture at mid-levels which accelerates the upward motion (Peterson et al. 2017; Fromm et al., 2019). Evidence exists that absorption of solar radiation can cause smoke to self-loft even higher (de Laat et al., 2012; Khaykin et al., 2020; Yu et al., 2019). The amount of smoke injected by pyroCb activity into the UTLS is comparable to volcanic levels (Peterson et al., 2018). Recent evidence also suggests that self-lofting caused smoke from Siberian wildfires to enter the UTLS in the summer/autumn of 2019 without the need for pyroconvection (Ohneiser et al., 2021), though this

possibility is still under investigation among the community (e.g., Boone et al., 2022). Finally, ammonium nitrate particles can reach the UTLS within the Asian Tropopause Aerosol Layer by way of convection in the Asian Monsoon region (Vernier et al., 2018; Höpfner et al. 2019).

The Cloud Aerosol Lidar with Orthogonal Polarization (CALIOP) onboard the Cloud-Aerosol Lidar and Infrared Pathfinder Satellite Observation (CALIPSO) platform was launched in April 2006 and is well suited to observe aerosol in the

lower stratosphere (Winker et al., 2009; Vernier et al., 2009). Being a vertical profiling lidar operating at two wavelengths (532 and 1064 nm), CALIOP can measure plume altitudes with high precision; data are reported at vertical resolutions of 60–300 m for stratospheric altitudes (Hunt et al., 2009). Depolarization ratio measurements at 532 nm also provide critical information on particle shape, of which the CALIOP cloud-aerosol discrimination and aerosol subtyping algorithms take full advantage (Kim et al., 2018; Liu et al., 2019). There are two important reasons for CALIOP level 2 algorithms to accurately

discriminate between stratospheric aerosol types. First, identifying different aerosol types allows researchers to characterize the abundance of these types and to quantify their disparate radiative, chemical, and dynamical impacts – the scientific motivation. Second, the CALIOP retrieval scheme relies on its aerosol subtyping algorithm to select an appropriate lidar ratio, which is required to accurately retrieve extinction and to correct for overlying attenuation when retrieving optical properties of underlying layers – the algorithmic motivation.

Given these compelling motivations, the CALIPSO project included new stratospheric aerosol subtypes for aerosol layers detected above the tropopause in the 2016 release of the version 4.1 level 2 data products (V4.1). These subtypes



included volcanic ash, sulfate/other, smoke, and polar stratospheric aerosol (PSA). Details of the initial stratospheric aerosol subtyping algorithm are given in Kim et al. (2018). In short, the algorithm discriminated between volcanic ash, smoke, and "sulfate/other" based on empirically derived thresholds of estimated particulate depolarization ratio at 532 nm and total attenuated backscatter color ratio (the ratio of attenuated backscatters at 1064 nm and 532 nm). Volcanic ash particles, aspherical in nature, exhibit a higher depolarization signature than sulfate aerosols which are spherical (Pueschel 1996), yielding low depolarization ratios (Vernier et al., 2013). Smoke observed in the stratosphere from pyroCb events can also be depolarizing (Haarig et al., 2018; Ohneiser et al., 2020), but to a lesser degree than ash. Layers with low 532 nm integrated attenuated backscatter (i.e., less than 0.001 sr$^{-1}$) were classified as sulfate/other. These tenuous layers represent the "other" fraction of the combined class. They were combined with sulfates under the assumption that the long residence time of sulfate aerosol would eventually yield low attenuated backscatter returns. Meanwhile, layers classified as aerosol at exceptionally low temperatures over the polar regions during polar stratospheric cloud (PSC) season were classified as PSA.

Since the initial release of the V4.1 level 2 data products, it became clear that some refinements were necessary. One significant example is the poor accuracy of discriminating between volcanic ash and smoke. During the massive Pacific Northwest (PNW) event in August 2017 (Peterson et al., 2018), smoke injected into the lower stratosphere was highly depolarizing (Haarig et al., 2018), exceeding the depolarization threshold used by the V4.1 algorithm to separate smoke and ash. As a consequence, over 58 % of the aerosol layers detected from this event were misclassified as volcanic ash. An example CALIOP observation of one of the earliest smoke plumes detected from the event (Torres et al., 2020) is shown in Fig. 1, illustrating the dominance of ash misclassification.

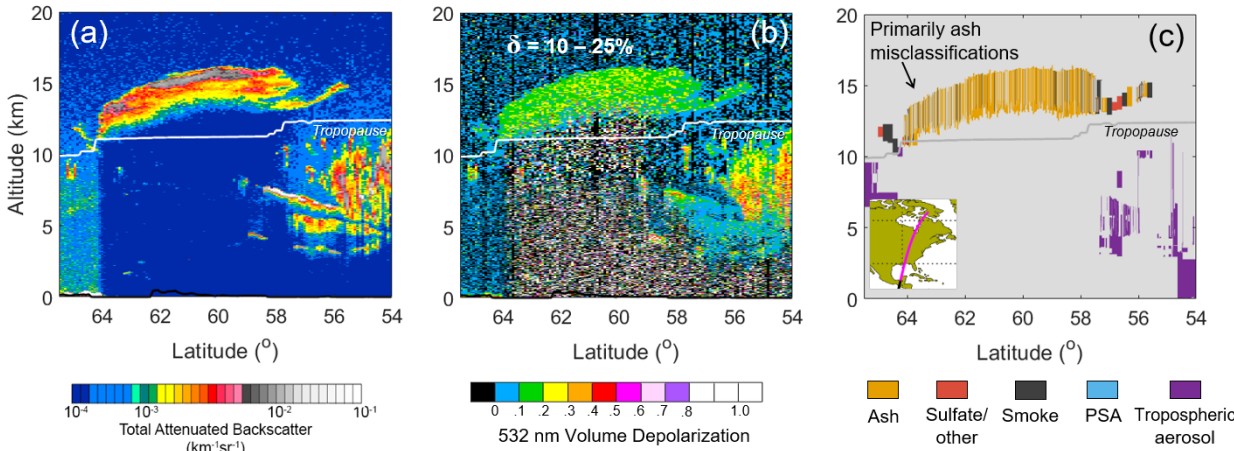

**Figure 1.** PyroCb smoke plume from the PNW event on 17 August 2017 at ~8:40 UTC over Quebec Province, Canada: (a) 532 nm total attenuated backscatter, (b) 532 nm volume depolarization ratio, and (c) V4.1 aerosol subtype classification from the level 2 vertical feature mask. Inset map shows the CALIOP ground track.





Another area needing refinement was the sulfate/other class. This subtype is ambiguous for practical applications because it is shared both by layers that could legitimately be sulfate and by weakly scattering layers that could be any aerosol type. In addition, the 532 nm integrated attenuated backscatter ($\gamma'_{532}$) threshold used to identify these weakly scattering layers was too high for the features commonly detected in the stratosphere by CALIOP. Some 75 % of all stratospheric

aerosol layers detected in V4.1 were classified as sulfate/other due to this threshold. For example, Fig. 2 shows several plumes from the June 2011 Puyehue-Cordón Caulle eruption where the layer depolarization ratios are elevated, indicating the presence of ash. However, some portions are classified as sulfate/other because their integrated attenuated backscatter falls below the threshold. A more reasonable classification for these layers would be ash.

Finally, studies have shown that the most common lidar ratio for volcanic ash may be larger than the default lidar

ratio used for the ash subtype in V4.1 (e.g., Prata et al., 2017). Based on these observations, the stratospheric aerosol subtyping algorithm and lidar ratio assignments have been updated for the V4.5 level 2 data release.

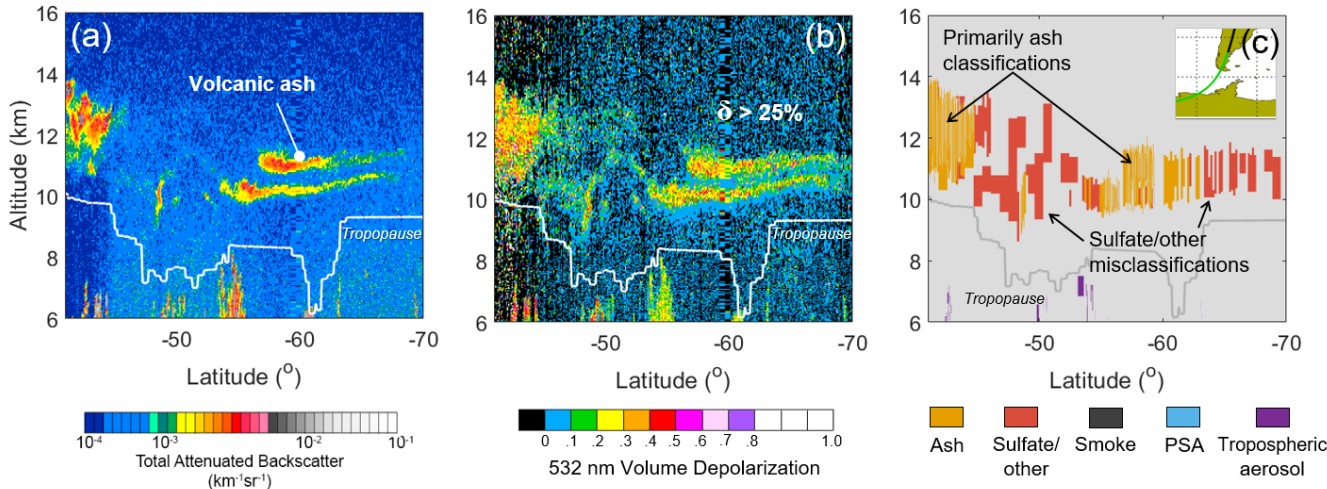

**Figure 2.** Volcanic ash plume from the June 2011 Puyehue-Cordón Caulle eruption observed on 15 June 2011 at ~5:42 UTC, southwest of
Argentina: (a) 532 nm total attenuated backscatter, (b) 532 nm volume depolarization ratio, and (c) V4.1 aerosol subtype classification from the level 2 vertical feature mask. Inset map shows the CALIOP ground track.

The organization of this paper is as follows. First, we describe the datasets used for our analysis in Sect. 2. Next, we review the V4.1/V4.2 stratospheric aerosol subtyping algorithm in Sect. 3 and then in Sect. 4 we describe the changes

implemented to the algorithm for V4.5. We provide a statistical summary of layer classifications between the versions in Sect. 4.6 to demonstrate the improvements of the revised algorithm. Section 5 is a performance assessment for select events dominated by volcanic ash, sulfate, and smoke to explore the classification fidelity. Concluding remarks are given in Sect. 6.



## 2 Data used

The data used for our analysis is from CALIOP onboard CALIPSO which has been operating since June 2006. CALIOP is
an elastic backscatter lidar measuring vertical profiles of attenuated backscatter at 532 nm and 1064 nm, with depolarization
capability at 532 nm (Hunt et al., 2009; Winker et al., 2009). Level 2 retrievals detect features using a threshold-based
algorithm after averaging the lidar backscatter signal to multiple horizontal resolutions (Vaughan et al., 2009). The primary
horizontal averaging resolutions for aerosol layers are 5 km, 20 km, and 80 km. Following detection, the cloud-aerosol
discrimination (CAD) algorithm classifies each layer as either cloud or aerosol based on the layer geolocation and measured
optical properties (Liu et al., 2019). Layers classified as aerosol are further classified as either tropospheric aerosol or
stratospheric aerosol, depending on their altitudes with respect to the tropopause, which is obtained from the Modern-Era
Retrospective analysis for Research Applications, Version 2 (MERRA-2) meteorological reanalysis product (Gelaro et al.,
2017). Specifically, aerosol layers having a 532 nm attenuated backscatter centroid below the tropopause are identified as
one of seven different tropospheric aerosol subtypes: clean marine, dust, polluted continental/smoke, clean continental,
polluted dust, elevated smoke, or dusty marine (Kim et al., 2018). Aerosol layers having a centroid above the tropopause are
assigned to one of the stratospheric aerosol subtypes that are the subject of this paper.

CALIOP level 2 retrievals are reported in a variety of data products. The stratospheric aerosol subtyping algorithm
is derived from layer properties reported in the level 2 aerosol layer product, and it is this same product that we primarily use
to evaluate the performance of the algorithm. The level 2 vertical feature mask, level 2 aerosol profile, and level 1B products
are also used for demonstrating individual case studies.

To characterize the performance of the previous V4.1 stratospheric aerosol subtyping algorithm, we use the V4.2
level 2 data products. The V4.1 level 2 data products were replaced with V4.2 in 2018 to add a new science data set (SDS)
reporting the minimum laser energy in each 80-km horizontal segment (the fundamental level 2 processing interval). This
new SDS was added to assist in quality screening data affected by low laser energy shots which began occurring primarily
over the South Atlantic Anomaly (SAA) region in mid-2016 (CALIPSO Data Advisory Page, 2018). To reliably exclude
affected level 2 retrievals in our analysis of V4.1 and V4.5 data, we use this SDS to impose the requirement that all laser
pulses have at least 60 mJ within any 80-km horizontal segment. Note that beyond adding the new SDS, there is no
difference between V4.1 and V4.2 in these level 2 products. In particular, the stratospheric aerosol subtyping algorithm is
identical in both data releases. Because V4.1 level 2 data is no longer publicly distributed, for the remainder of the paper we
will refer only to V4.2.

To assess the performance of the revised algorithm, we use a pre-release test version of the V4.5 level 2 products. This test version accurately reflects the behavior of stratospheric aerosol subtyping in the official V4.5 level 2 release, planned for 2022. Updates to the V4.5 level 2 algorithms – beyond those described in this paper – are primarily related to tropospheric feature classification, boundary layer cloud clearing, and optical depth retrievals that are fully independent of
aerosol subtyping (Tackett et al., 2021, 2022; Ryan et al., 2022). These updates have no impact on our analysis of



stratospheric aerosol detections. The V4.51 level 1B data used as input for the V4.5 level 2 products also contains several updates relative to the previous release (CALIPSO Lidar Level 1 V4.51 Data Quality Statement, 2022). The 532 nm daytime and 1064 nm calibration algorithms now mitigate the influence of low energy laser shots on the derived calibration coefficients. This corrects biases and reduces calibration uncertainty in these channels at SAA latitudes (~15° S to 30° S)
since the onset of low energy shots in mid-2016. Small corrections have been made to the 1064 nm baseline shape having negligible impacts on the current analysis. Lastly, adjustments have been made to the polarization gain ratios to properly account for day and night differences rather than use the same value regardless of lighting conditions. The primary impact of these adjustments is to increase nighttime depolarization ratios by ~4 % and decrease daytime depolarization ratios by ~1 %.

As a final note, we emphasize that the "layers" discussed throughout this paper are those detected by CALIOP at 5
km, 20 km, and 80 km resolutions. Each layer is only counted once regardless of horizontal extent because our intention is to characterize the classification frequencies for the unique layers that are input to the subtyping algorithm. As such, the true geospatial extent of aerosol from each event is not explicitly represented.

## 3 Summary of the V4.2 stratospheric aerosol subtyping algorithm

The V4.2 stratospheric aerosol subtyping algorithm, documented by Kim et al. (2018), evaluates several CALIOP
measurables to identify four subtypes: volcanic ash, sulfate/other, smoke, and polar stratospheric aerosol (PSA). The method for discriminating between volcanic ash, sulfate, and smoke is based on an empirical analysis of joint distributions of estimated particulate depolarization ratio and feature integrated attenuated backscatter color ratio derived from the level 2 aerosol layer product for manually classified aerosol layers from specific events where the dominant aerosol type is well documented in the literature. Whereas manual classification is far more error-prone in the troposphere where multiple
aerosol types often coexist, stratospheric aerosol events such as volcanic eruptions and large pyroCb injections tend to be episodic and their compositions are characterized by various independent methods. The events selected for the joint distributions are the only significant contributors to stratospheric aerosol loading at the time they are sampled, typically during the first 30 days after event initiation. Plumes are tracked manually in CALIOP imagery over successive days and their latitude/longitude/altitude boundaries are recorded for each CALIOP granule. In order to avoid cloud contamination,
plumes near or in contact with high altitude cirrus or overshooting cloud tops are excluded. All layers detected in the level 2 aerosol layer product within the plume boundaries contribute to the joint distributions. The full list of CALIOP granules and plume boundaries for all events analyzed for V4.2 and V4.5 development is reported in the Supplement.

The first dimension of the joint distribution relies on the depolarization sensitivity of CALIOP. An elevated depolarization ratio is an excellent discriminator for identifying non-spherical particles such as volcanic ash, dust, and cirrus
relative to spherical particles such as sulfate aerosol. Volume depolarization ratio ($\delta_v$) is calculated from the ratio of CALIOP measurements of attenuated backscatter ($\beta'$) measured perpendicular and parallel to the plane of the emitted pulse, which was originally linearly polarized:



$$\delta_v = \frac{\beta'_\perp}{\beta'_\parallel} \tag{1}$$


The $\delta_v$ contains contributions from both molecular and particulate scattering, which can under-represent the particulate depolarization of weakly scattering features. In order to correct for the molecular contribution, CALIOP aerosol subtyping uses an estimated particulate depolarization ratio ($\delta_p^{est}$) according to Eq. (2) (Omar et al., 2009). Here, the mean attenuated scattering ratio (Kar et al., 2018) is defined as $R_{mas} = \langle \beta'(z)/\beta'_{mol}(z) \rangle$, where the angle brackets indicate averaging over the vertical extent of a layer. The molecular attenuated backscatter, $\beta'_{mol}(z)$, is computed from the MERRA-2 model and the molecular depolarization ratio, $\delta_m$, is 0.00366 (Hostetler et al., 2006).

$$\delta_p^{est} = \frac{\delta_v \left[ \left( R_{mas} - 1 \right) \left( 1 + \delta_m \right) + 1 \right] - \delta_m}{\left( R_{mas} - 1 \right) \left( 1 + \delta_m \right) + \delta_m - \delta_v} \tag{2}$$


The second dimension of the joint distribution is feature integrated attenuated backscatter color ratio, $\chi'$, which can give qualitative information about particle size. This quantity is ratio of the feature integrated attenuated backscatters ($\gamma'_\lambda$), computed separately at both lidar wavelengths between layer top and base, and is derived as follows:

$$B_{\lambda,k} = \frac{\beta'_\lambda(z_k)}{T^2_{m,\lambda}(z_k) T^2_{O_3,\lambda}(z_k)} \tag{3}$$

$$g_\lambda = \frac{1}{2} \sum_{k=top+1}^{base} \left( z_{k-1} - z_k \right) \left( B_{\lambda,k-1} + B_{\lambda,k} \right) \tag{4}$$

$$\gamma'_\lambda = g_\lambda - \left[ \frac{1}{2} \left( z_{top} - z_{base} \right) \left( B_{\lambda,top} + B_{\lambda,base} \right) \right] \tag{5}$$

$$\chi' = \gamma'_{1064} / \gamma'_{532} \tag{6}$$

This formulation corrects the attenuated backscatter coefficients for molecular and ozone attenuation and applies an approximate correction for molecular scattering (Vaughan et al., 2005), which is important for low optical depth layers typically found in the stratosphere. Molecular and ozone two-way transmittances $T^2_m$ and $T^2_{O_3}$ are obtained from the MERRA-2 model.




The joint distributions used for the V4.2 release are shown in Fig. 3 and the contributing events are documented in Table 1. Note that only a subset of the events in Table 1 contributed to the V4.2 analysis; more events were added for V4.5

(discussed in Sect. 4). The contributions for volcanic ash and sulfate in V4.2 were dominated by the June 2011 Puyehue-Cordón Caulle eruption and the August 2008 Kasatochi eruption, respectively. Very few stratospheric smoke events had been observed when these distributions were first constructed in 2015 during development for the V4.2 release. The primary contributor was the February 2009 "Black Saturday" bushfires in Australia where smoke reached altitudes of 19 km in the southern mid-latitudes (Siddaway and Petelina, 2011). In order to more fully sample $\delta_p^{est}$ and $\chi'$ for smoke, several high-

altitude, yet not stratospheric, smoke events were included in the joint distributions. From this, two populations emerged: smoke with low depolarization in the troposphere and higher depolarization in the stratosphere (Fig. 3). As we shall discuss later, enhanced depolarization is a common feature of smoke reaching the stratosphere associated with pyroCb activity, though we had not fully appreciated this fact during the V4.2 development.

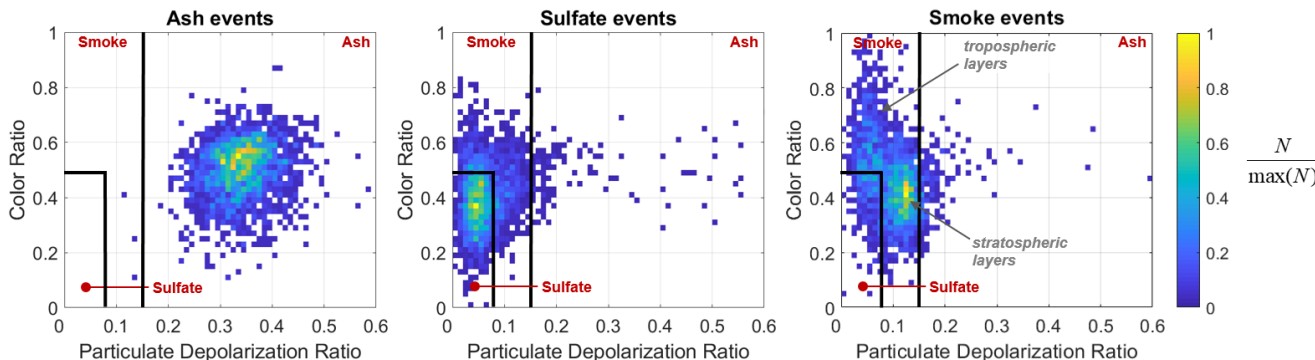


**Figure 3.** Joint distributions used in V4.2 development: estimated particulate depolarization ratio ( $\delta_p^{est}$ ) and feature integrated attenuated backscatter color ratio ( $\chi'$ ) for manually classified layers during events dominated by the aerosol type indicated in the title. Only layers with $\gamma'_{532} > 0.001$ sr$^{-1}$ contribute. Histograms of layer numbers ($N$) are min-max normalized. Black lines indicate discrimination thresholds in V4.2 and red text indicates the algorithm classification.


Using the joint distributions in Fig. 3, thresholds were established for ash, sulfate, and smoke that minimized their overlap in measurement space. The flowchart in Fig. 4 shows how these thresholds are used in the V4.2 release. Stratospheric aerosol layers having $\delta_p^{est} > 0.15$ are classified as ash. Separate branches exist to capture smoke with low depolarization ( $\delta_p^{est} < 0.075$ ) and with high depolarization ($0.075 < \delta_p^{est} < 0.15$ ), depending on the layer color ratio. Sulfate

is identified for layers having low $\delta_p^{est}$ and low $\chi'$ .

Note that the name given to the V4.2 subtype containing sulfate is "sulfate/other". The "other" contributions to this subtype are layers having low integrated attenuated backscatter. The ability to accurately discriminate between aerosol types





using depolarization and color ratio requires measurements having sufficient backscatter magnitudes. This is especially relevant in the stratosphere where exceptionally low optical depths are common. Therefore, a test was placed prior to the

determination of ash, sulfate, or smoke in the V4.2 flowchart (Fig. 4) to weed out weakly scattering features using the feature integrated attenuated backscatter at 532 nm ($\gamma'_{532}$) defined by Eq. (5).

In the V4.2 data release, all stratospheric aerosol layers having $\gamma'_{532} < 0.001$ sr$^{-1}$ (hereafter named "low-$\gamma'$ layers") that were not previously classified as PSA are assigned to the sulfate/other subtype. The rationale for combining sulfate and low-$\gamma'$ layers was because we assumed volcanic sulfate would eventually become weakly scattering over time since this

aerosol type tends to persist in the stratosphere for weeks to months after injection, all-the-while becoming increasingly diffuse and hence decreasing in optical depth.

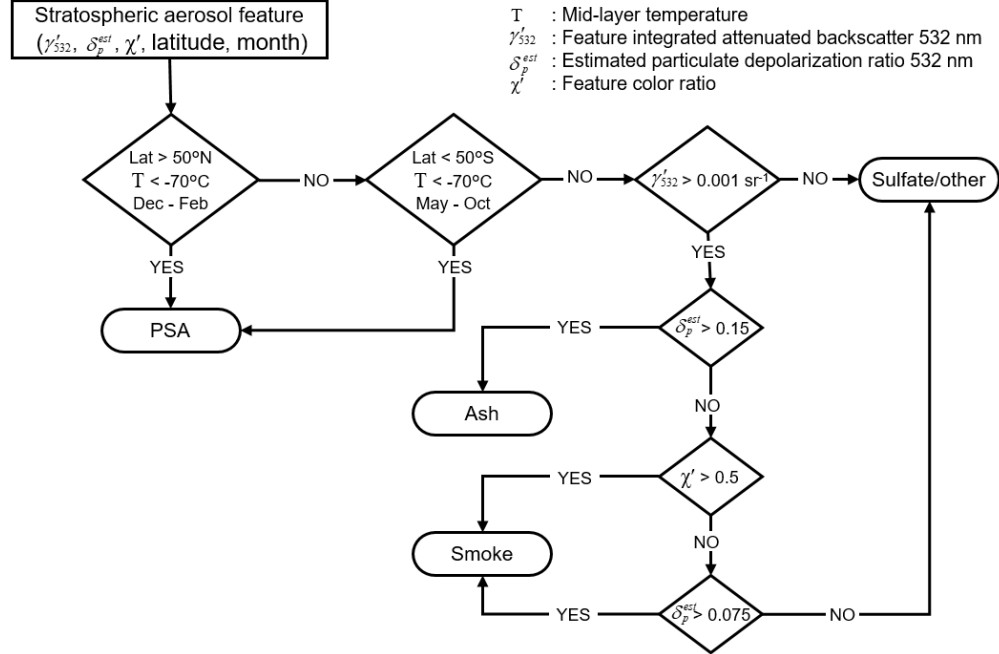

**Figure 4.** Flowchart for stratospheric aerosol subtyping in V4.2.

The fourth subtype given by the stratospheric aerosol subtyping algorithm is for polar stratospheric aerosol. The PSA type is meant to assign a reasonable classification for aerosol layers identified in regions where PSC formation is likely. The method to identify PSA is identical for both the V4.2 and V4.5 releases, so it is only briefly summarized here. PSA layers are identified first by evaluating the layer midpoint temperature for stratospheric aerosol layers during polar-winter

PSC seasons. Based on the climatology of Poole and Pitts (1994), PSC season is assumed to be December to February for



the Arctic and May to October for the Antarctic. A latitude threshold of 50° confines PSA classifications to the appropriate pole and encompasses the expected range of PSC/PSA formation. Temperature information is provided by the MERRA-2 model. The −70 °C layer midpoint temperature maximum is based on the midpoint temperatures of stratospheric aerosol layers detected by CALIOP in these regions/seasons (see Fig. 5 of Kim et al. (2018)). It also agrees with the observed

temperatures for PSC formation of Rosen et al. (1997).

The classification of PSA is the first decision in the stratospheric aerosol typing flowchart in Fig. 4. This step confines the PSA classification to only those geographical regions and seasons where they are expected to exist. These layers are often detected adjacent to features classified as cloud and have elevated depolarization. It is quite possible that some fraction of these are CAD misclassifications along the fringes of clouds rather than legitimate aerosol (Liu et al., 2019). The

PSA classification prevents these layers from being misclassified as volcanic ash when none exists (both having elevated depolarization). The flip side of this is that when true volcanic ash enters these regions during PSC season, it may not be classified correctly. An example is considered in Sect. 5.1. The confidence in the PSA classification is considered low at this time and its accuracy is not evaluated further in this paper. Users investigating PSC observations by CALIOP are instead referred to the level 2 polar stratospheric cloud mask product which is specialized for this purpose

(NASA/LARC/SD/ASDC, 2016a; Pitts et al., 2018).

We now turn our attention to the changes made to the stratospheric aerosol subtyping algorithm for the V4.5 release and how they improve the shortcomings in the previous V4.2 release.

## 4 The version 4.5 stratospheric aerosol subtyping algorithm

In Sect. 1 we highlighted the need to improve several aspects of the stratospheric aerosol subtyping algorithm. The goals of

the revisions implemented in the V4.5 are therefore to:

- Improve discrimination between volcanic ash and smoke in the stratosphere.
- Remove the ambiguity in the sulfate/other subtype.
- Reduce the number of layers classified as low-$\gamma'$.
- Update the volcanic ash lidar ratio.

In order to accomplish this, two structural changes were made to the algorithm and two thresholds were adjusted. The new flowchart for V4.5 is shown in Fig. 5. As in the previous release, discrimination between volcanic ash, sulfate, and smoke is determined by an empirical analysis of joint distributions of depolarization and color ratios from events where the

aerosol type is known. The joint distributions used for the V4.5 analysis are shown in Fig. 6. More events have been added since the V4.2 analysis so that we can better characterize the depolarization and color ratio for these types (denoted by daggers in Table 1). The new events include two volcanic eruptions: the sulfate layers from the 2009 Sarychev Peak eruption and ash layers from the 2015 Mount Calbuco eruption. Their composition was determined empirically based on elevated





depolarization. The change in smoke layers sampled for the new distribution had a critical influence on the revisions that were made. In particular, smoke layers from the August 2017 Pacific Northwest (PNW) event and the December 2019/January 2020 Australian New Year (ANY) event were added. These events injected large amounts of depolarizing smoke into the stratosphere (Peterson et al., 2018; Hu et al., 2019; Ohneiser et al., 2020). Stratospheric smoke layers were also added from the December 2006 Australian bushfires (Dirksen et al., 2009) and from the July 2014 North American wildfires. The much larger number of stratospheric smoke layers now contributing to the joint distribution analysis allows us

to exclude the tropospheric smoke layers that previously influenced the V4.2 threshold selections.

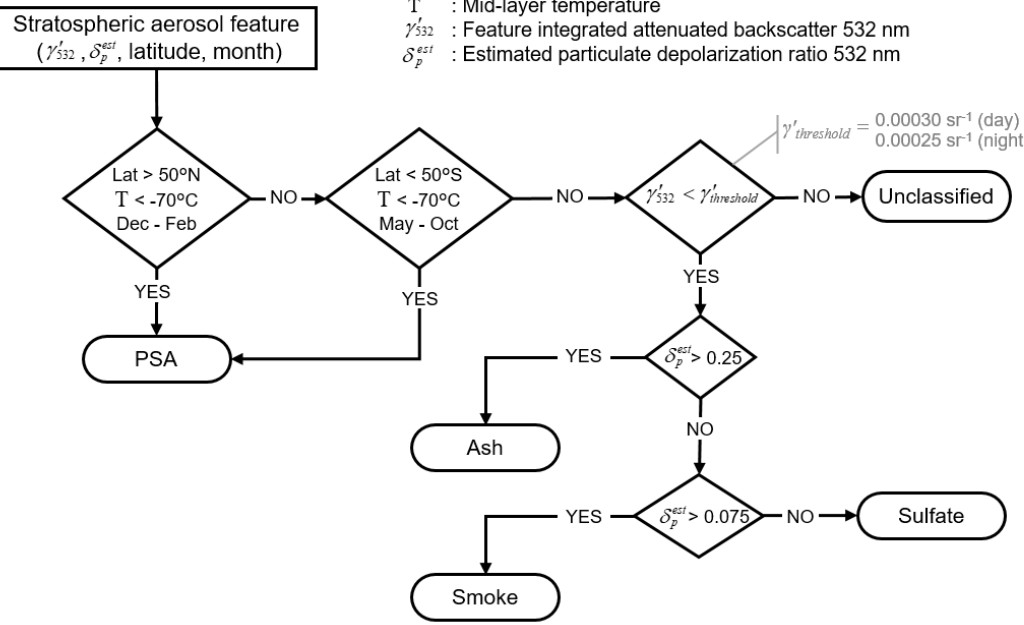

**Figure 5.** Flowchart for stratospheric aerosol subtyping in the V4.5.





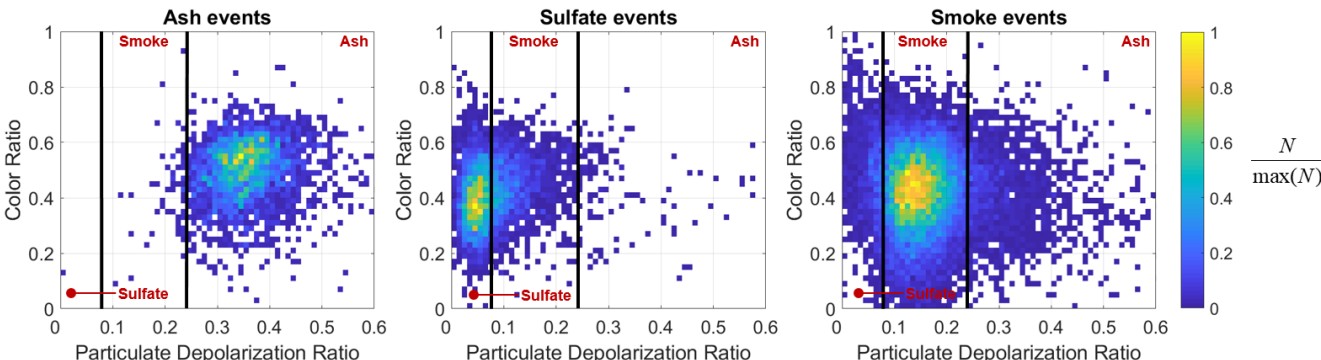


**Figure 6.** Joint distributions used in V4.5 development; same description as Fig. 3, but only stratospheric aerosol layers with $\gamma'_{532} > 0.001$ $sr^{-1}$ contribute. Black lines indicate discrimination thresholds in V4.5 and red text indicates the algorithm classification.

**Table 1.** Manually classified events used to establish thresholds to discriminate between volcanic ash, volcanic sulfate, and smoke in V4.5.
The number of unique stratospheric aerosol layers detected in V4.5 is given along with the fraction of these layers in night granules, and the dominant aerosol type. Only layers with $\gamma'_{532} > 0.001$ $sr^{-1}$ contribute. The individual CALIOP granule names and latitude/longitude/altitude information for these layers are given in the Supplement.

| N layers in V4.5 | Fraction at night (%) | Event | Dominant aerosol type |
|---|---|---|---|
| 2528 | 100 | Puyehue-Cordón Caulle eruption, June 2011 | Volcanic ash (Bignami et al., 2014) |
| 687 | 50 | †Calbuco eruption, April 2015 | Volcanic ash (Marzano et al., 2018) |
| 36 | 100 | Chaitén eruption, May 2008 | Volcanic ash (Prata et al., 2010) |
| 3095 | 29 | †Sarychev eruption, June 2009 | Sulfate (Prata et al., 2017) |
| 2148 | 100 | Kasatochi eruption, August 2008 | Sulfate (Krotkov et al., 2010) |
| 186 | 100 | Nabro eruption, June 2011 | Sulfate (Theys et al., 2013) |
| 63 | 100 | Okmok eruption, July 2008 | Sulfate (Prata et al., 2010) |
| 6918 | 75 | †Australian New Year (ANY) event, Dec. 2019/Jan. 2020 | Smoke (Kablick et al., 2020) |
| 5016 | 63 | †Pacific Northwest (PNW) event, August 2017 | Smoke (Peterson et al., 2018) |
| 2177 | 11 | †North American wildfires, July 2014 | Smoke |
| 1720 | 63 | †Australian bushfires, December 2006 | Smoke (Dirksen et al., 2009) |
| 760 | 100 | Black Saturday Australian bushfires, February 2009 | Smoke (Siddaway and Petelina, 2011) |
| 187 | 100 | Siberian wildfires, May 2012 | Smoke |
| 15 | 100 | Siberian wildfires, June 2007 | Smoke |
| 0 | 0 | *Canadian wildfires, July–August 2007 | Smoke |

†New events added since the V4.2 development of the stratospheric aerosol typing algorithm described in Kim et al., 2018.

* Event is exclusively comprised of tropospheric layers used in V4.2 development, but not V4.5.



## 4.1 Color ratio test for smoke removed

The new joint distributions in Fig. 6 show that excluding tropospheric smoke layers from the sample population had an
important impact: the population of smoke with low depolarization and higher color ratio is no longer prominent (cf. Fig. 3).
Recent literature has confirmed the enhanced depolarization from smoke lofted by pyroCb events. The analysis of Christian
et al. (2020) demonstrated increased depolarization and decreased color ratio for fresh pyroCb plumes at high altitudes
compared to lower altitudes. This is corroborated by Fig. 7, which shows higher values of $\delta_p^{est}$ for stratospheric smoke layers
used for the V4.5 joint distribution compared to the tropospheric smoke layers contributing to the V4.2 joint distribution.


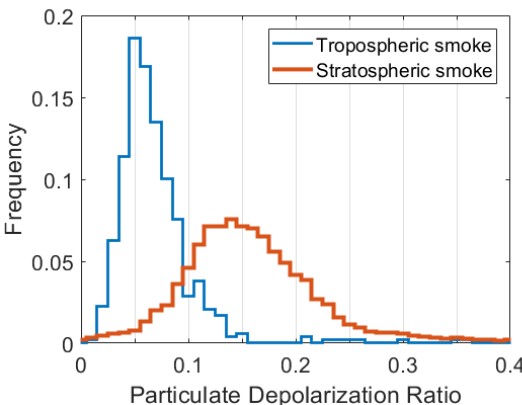

**Figure 7.** Frequency distributions of estimated particulate depolarization ratio ($\delta_p^{est}$) for stratospheric smoke layers in Table 1 (red) and
tropospheric smoke layers contributing to the V4.2 joint distributions (496 layers from the May 2012 Siberian wildfires and 36 layers from
July–August 2007 Canadian wildfires). Only layers with $\gamma'_{532} > 0.001$ sr$^{-1}$ contribute.

These observations (primarily representing pyroCb events) suggest that smoke injected to extremely high altitudes
contain particles that are aspherical and smaller compared to those injected to lower altitudes. The cause of aspherical
particles in smoke plumes from these pyroCb events is an active area of research (e.g., Gialitaki et al., 2020). However, the
message is clear. Smoke reaching the stratosphere in these events typically depolarizes the CALIOP backscatter signal more
than smoke that is confined to the troposphere. Based on this information, we removed the color ratio test, which previously
was employed to capture the low depolarization, higher color ratio smoke that is characteristic of tropospheric events. This
marks the first structural change. The V4.5 stratospheric aerosol subtyping algorithm now strictly relies on depolarization
ratio to discriminate between ash, sulfate, and smoke. Removing the color ratio test has the additional effect of allowing
more sulfate layers to be classified correctly as sulfate rather than smoke.

There is a possibility of misclassification by relying solely on depolarization ratio to discriminate between sulfate
and smoke. Recent research by Ohneiser et al. (2021) hypothesizes that smoke from Siberian wildfires in 2019 self-lofted





from the troposphere into the UTLS. Because this smoke was aged and not of pyroCb origin, its depolarization was low (<

0.05) causing the CALIOP V4.2 stratospheric aerosol subtyping algorithm to misclassify these smoke layers as sulfate

(Ansmann et al., 2021). The subtyping algorithm will continue to struggle in these cases in V4.5 due to the similarly low

depolarization ratios of tropospheric smoke and sulfate. It is currently unknown how often smoke plumes reach the UTLS

exclusively by self-lifting.

## 4.2 Depolarization ratio threshold between smoke and ash increased

The estimated particulate depolarization ratio ($\delta_p^{est}$) threshold to discriminate between ash and smoke was 0.15 in the V4.2

release. This threshold worked well for the 2009 Black Saturday Australian bushfires which had particulate depolarization

ratios around 0.10 to 0.15. However, depolarization ratios were higher for stratospheric smoke layers from the PNW event in

August 2017 (Fig. 8a). European lidar systems observed 532 nm particulate depolarization ratios ranging from 0.15 to 0.2 in

the two weeks following the event (Ansmann et al, 2018; Haarig et al., 2018; Khaykin et al., 2018; Hu et al., 2019). Figure

8b shows the distributions of CALIOP estimated particulate depolarization for all manually classified stratospheric smoke

and volcanic ash events listed in Table 1. The markedly larger depolarization of layers from the PNW event compared to

previous stratospheric smoke events caused a large frequency of misclassification in V4.2: whereas ~25 % of all smoke

events excluding the PNW event are misclassified as volcanic ash with the 0.15 threshold, a whopping 58 % of the smoke

layers detected during the PNW event were misclassified as ash.


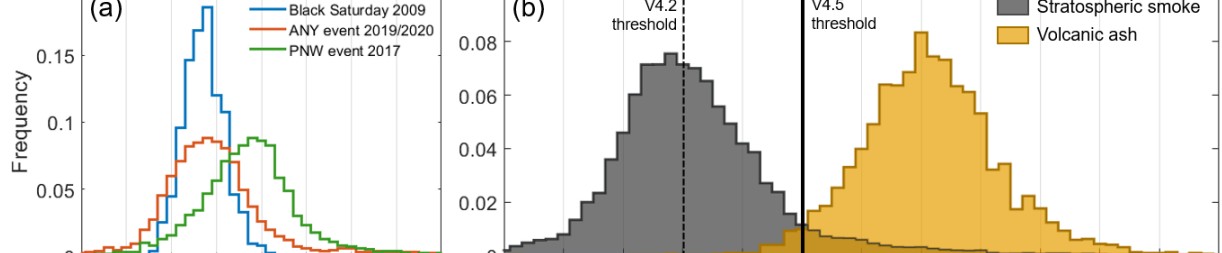

**Figure 8.** Frequency distributions of estimated particulate depolarization ratio ($\delta_p^{est}$) for layers identified in Table 1, comparing (a) three
major stratospheric smoke events and (b) all stratospheric smoke events to all volcanic ash events. Only layers with $\gamma'_{532}$ > 0.001 sr$^{-1}$
contribute.

In order to better discriminate between stratospheric smoke and volcanic ash in V4.5, the $\delta_p^{est}$ threshold between

these types was increased from 0.15 to 0.25. This is roughly the minimum overlap between the distributions in Fig. 8b.





Doing so reduces the amount of smoke layers misclassified as ash in the PNW event to 9 % (Sect. 4.6). The example from this event in Fig. 9a shows that the dominant classification is now smoke rather than ash (cf. Fig. 1). A more thorough

assessment of the classification performance is given in Sect. 5.

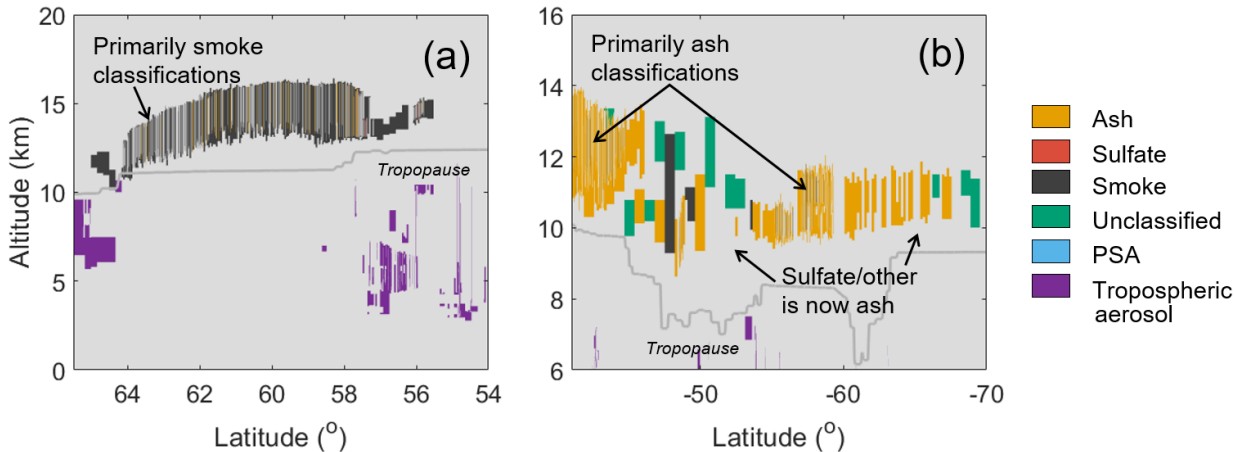

**Figure 9.** Aerosol subtyping after V4.5 revisions for (a) pyroCb smoke plume from the PNW event (cf. Fig. 1) and (b) volcanic ash plume from the June 2011 Puyehue-Cordón Caulle eruption (cf. Fig. 2).

**4.3 Sulfate/other separated into two types: sulfate and unclassified**

The second structural change to the stratospheric aerosol subtyping algorithm is the separation of the "sulfate/other" classification into two separate types: sulfate and "unclassified". Previously, both low-$\gamma'$ layers and layers meeting the sulfate criteria were given the "sulfate/other" classification. This hampered the ability to easily identify volcanic sulfate layers because low-$\gamma'$ layers can be any aerosol type, provided $\gamma'_{532}$ is sufficiently low. Returning to the example in the introduction,

Fig. 9b shows that the majority of the ash plume is now correctly classified whereas before it was classified as sulfate/other (cf. Fig. 2). This is a result of separating the low-$\gamma'$ branch and also lowering the $\gamma'_{532}$ threshold (discussed next). The nomenclature for the low-$\gamma'$ branch has also changed to "unclassified" to emphasize that no attempt has been made to classify the subtype based on its depolarization. This subtype serves as a catch-all for aerosol having insufficient backscatter to yield confident classifications of volcanic ash, sulfate, or smoke. We will characterize unclassified layers in Sect. 5.4.

**4.4 Low-$\gamma'$ threshold decreased**

The second threshold adjustment was for low-$\gamma'$ layers, now identified as "unclassified". Previously the threshold was $\gamma'_{532} < 0.001$ sr$^{-1}$ which caused 75 % of all confidently classified stratospheric aerosol layers detected from June 2006– December 2018 to be classified as sulfate/other. In the V4.5 release, this threshold has been lowered to assign the unclassified subtype to layers having $\gamma'_{532}$ in the lowest quartile of the June 2006 – December 2018 stratospheric aerosol





distribution, corresponding to $\gamma'_{532} < 0.0003$ sr$^{-1}$ at day and $\gamma'_{532} < 0.00025$ sr$^{-1}$ at night. The lowest-quartile $\gamma'_{532}$ metric was selected so that the average relative uncertainty in $\delta^{est}_p$ is less than 250 % for layers classified as ash, sulfate, and smoke in this distribution at all detection resolutions – except for layers detected at 80-km resolution in daytime which typically have uncertainties of ~400 %. Classifications for those layers should be interpreted with caution. For reference, the average relative uncertainty in $\delta^{est}_p$ was less than 200 % for these classifications based on low-$\gamma'$ threshold in V4.2. With the new

thresholds, layers detected at 5 km and 20 km horizontal resolution, regarded as robustly scattering features, will almost certainly be subtyped as something other than unclassified, as will approximately half of the 80 km resolution layers. This reduces the frequency of unclassified aerosol layers to 25 % with a corresponding increase of 50 % in the relative uncertainty of $\delta^{est}_p$ which is not expected to noticeably degrade the fidelity of the subtyping algorithm. Rather, we consider the increase in opportunities to classify stratospheric aerosol layers outweighs the increase in uncertainty.

375        If we assume a lidar ratio of 50 sr for sulfates (i.e., as in Kim et al. (2018)), we can translate the $\gamma'_{532}$ thresholds into approximate optical depth thresholds. Using the V4.2 threshold of 0.001 sr$^{-1}$ caused all layers with optical depths less than ~0.053 to classified as sulfate/other. By contrast, the revised $\gamma'_{532}$ threshold of 0.0003 sr$^{-1}$ translates into an optical depth threshold of ~0.015, so that only those layers with optical depths less than ~0.015 are identified as unclassified.

### 4.5 Lidar ratio for ash increased

The stratospheric aerosol lidar ratios assignments for V4.5 are shown in Table 2. The same values were used in V4.2 for sulfate (previously sulfate/other), smoke, and PSA, as justified by Kim et al., 2018. The lidar ratio for the unclassified subtype is based on extinction retrieval considerations. As discussed in Sect. 4.3, unclassified layers can be any subtype provided their $\gamma'_{532}$ is sufficiently low. In order to reduce the impact of extinction retrieval errors that propagate into underlying layers, it is better to use a lidar ratio that is too low rather than too high when accurate knowledge of the subtype

is unavailable (Young et al., 2013). Therefore, the unclassified subtype shares the same lidar ratios as sulfate because these are the lowest lidar ratios expected for non-PSA stratospheric aerosol layers.

         The 532 nm lidar ratio for ash was increased in V4.5. In the previous release, the default lidar ratio for volcanic ash was set to 44 sr at both 532 nm and 1064 nm, matching the lidar ratios of the tropospheric dust subtype (Kim et al., 2018). This choice was motivated by similarities between the size distributions of volcanic ash and dust (Winker et al., 2012), and

to avoid discontinuities in extinction retrievals between ash above and below the tropopause (because ash and dust are depolarizing, volcanic ash will be misclassified as dust in the troposphere). However, several analyses suggest that the 532 nm lidar ratio for ash is higher. Ash plumes from the April 2010 Eyjafjallajökull eruption, observed by EARLINET lidar measurements, revealed 532 nm lidar ratios over Germany from 45–60 sr (Ansmann et al., 2010; Groß et al., 2012). Higher lidar ratios for the plume were observed over Italy, ranging from 50–92 sr (Mona et al., 2012), and over Greece, ranging

from 44 to 88 sr (Kokkalis et al., 2013). These latter studies suggest that relative humidity and/or ageing may have played a





role in the variability. Prata et al. (2017) used CALIOP constrained retrievals to characterize ash lidar ratios from the 2011 Puyehue-Cordón Caulle eruption. The constrained retrieval method computes transmittance if there is clear air above and below a feature, thus allowing a measurement of the layer lidar ratio. Their analysis reveals a median ash lidar ratio of 67 sr for this event.


**Table 2.** Lidar ratios for stratospheric aerosol subtypes in V4.5.

| Aerosol Subtype | $S_{532}$ (sr) | $S_{1064}$ (sr) |
|---|---|---|
| Volcanic ash | 61 ± 17 | 44 ± 13 |
| Sulfate | 50 ± 18 | 30 ± 14 |
| Smoke | 70 ± 16 | 30 ± 18 |
| Unclassified | 50 ± 18 | 30 ± 14 |
| Polar stratospheric aerosol | 50 ± 20 | 25 ± 10 |

Based on the growing consensus in the literature that the 532 nm lidar ratio for volcanic ash is typically larger than the value used in V4.2, we have revised the lidar ratio in the V4.5 release. Following the method of Prata et al. (2017), Fig.

10 shows the lidar ratios from CALIOP constrained retrievals for ash layers from several eruptions including the Puyehue-Cordón Caulle (2011), Kelud (2014), Sarychev (2009), and Calbuco (2015). In order to remove no-confidence retrievals, only layers with retrieved lidar ratio uncertainty < 100 % and |CAD score| > 20 contribute to the histogram. Using the mean and standard deviation of lidar ratios derived from this analysis, the default 532 nm lidar ratio is increased to 61 ± 17 sr, consistent with values reported in the literature. Because knowledge of 1064 nm lidar ratios is not as broad in the literature,

the default 1064 nm lidar ratio for ash will not be changed for the V4.5 release.

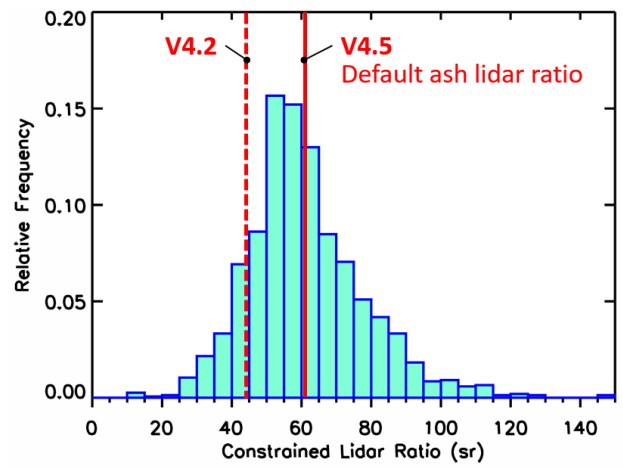



**Figure 10.** Distribution of 532 nm lidar ratios retrieved from constrained retrievals of ash layers from several volcanic events (see text). Lines indicate default values for ash in V4.2 and V4.5.

## 4.6 Change in classifications between V4.2 and V4.5

The changes in classifications due to V4.5 revisions are summarized in Fig. 11 based on all the manually classified layers in Table 1. The hatched bars indicate low-$\gamma'$ features ("other" in V4.2 and unclassified in V4.5). One obvious change for ash-dominant events is the radical reduction of the sulfate/other subtype and accompanying increase in ash classifications. This is a direct consequence of reducing the threshold for low-$\gamma'$ layers; the new ash layers have $\gamma'_{532}$ somewhere between the old and new qualifying thresholds. Evidently, $\delta_p^{est}$ is still an excellent discriminator for ash even as $\gamma'_{532}$ decreases, bolstering our confidence in our decision to reduce the $\gamma'_{532}$ threshold. There is a minor increase in smoke misclassifications where $\delta_p^{est}$ for some ash layers is just low enough to resemble that of depolarizing smoke.

For the sulfate-dominant events in V4.2, a substantial fraction of sulfate/other classifications were low-$\gamma'$ layers (hatched bars in Fig. 11). A small number of these low-$\gamma'$ layers became unclassified in V4.5 after separating the sulfate/other class. Meanwhile a larger number became classified as sulfate because reducing the $\gamma'_{532}$ threshold and removing the color ratio test allowed more opportunities for the sulfate classification. The net effect is that sulfate is the dominant subtype given for these events. There is a small increase in smoke misclassification for at least three reasons. These layers could be mixtures of sulfate and ash, yielding moderate values of $\delta_p^{est}$ (discussed further in Sect. 5.2). Another plausible explanation is the increased variability in $\delta_p^{est}$ for layers having $\gamma'_{532}$ in between the old and new low-$\gamma'$ thresholds allows more opportunities for sulfate layers to exceed the $\delta_p^{est} > 0.075$ threshold. Lastly, a small number of ash classifications changed to smoke due to the increase in $\delta_p^{est}$ threshold separating these types. Even though smoke misclassification increases for these sulfate-dominant events, eliminating the color ratio test improved the accuracy. For all the sulfate-dominant events shown in Fig. 11, the V4.5 classification frequency for sulfate and smoke is 70 % and 28 %, respectively. Retaining the color ratio test would have yielded classification frequencies of 58 % and 41 %, respectively. Given the limited CALIOP observables, discriminating sulfate from smoke will always be a challenge, and hence this seemingly modest improvement represents a welcome and useful increase in classification accuracy. There remains a substantial overlap in the $\delta_p^{est}$ distributions for these two types, which inherently reduces the ability to discriminate sulfate with a high degree of accuracy.

Classifications for pyroCb-smoke dominated events show a marked improvement. The frequency of layers classified as smoke increased while misclassifications as other types decreased. As previously mentioned, layers from the PNW event were primarily misclassified as ash in V4.2 (~58 %). Now smoke is the dominant classification (~85 %), with an ash misclassification rate around 9 %. A similar reduction in ash misclassification occurs in the other stratospheric smoke





events highlighted in Fig. 11. Lowering the low-γ′ threshold also improves smoke classification by moving many layers
previously classified as sulfate/other to the smoke subtype. Just as with volcanic ash, the elevated depolarization ratios
permit recognition of these layers as depolarizing smoke despite having lower $\gamma'_{532}$ There is a small increase in sulfate
misclassifications for smoke layers having $\delta_p^{est} < 0.075$ due to the removal of the color ratio test.

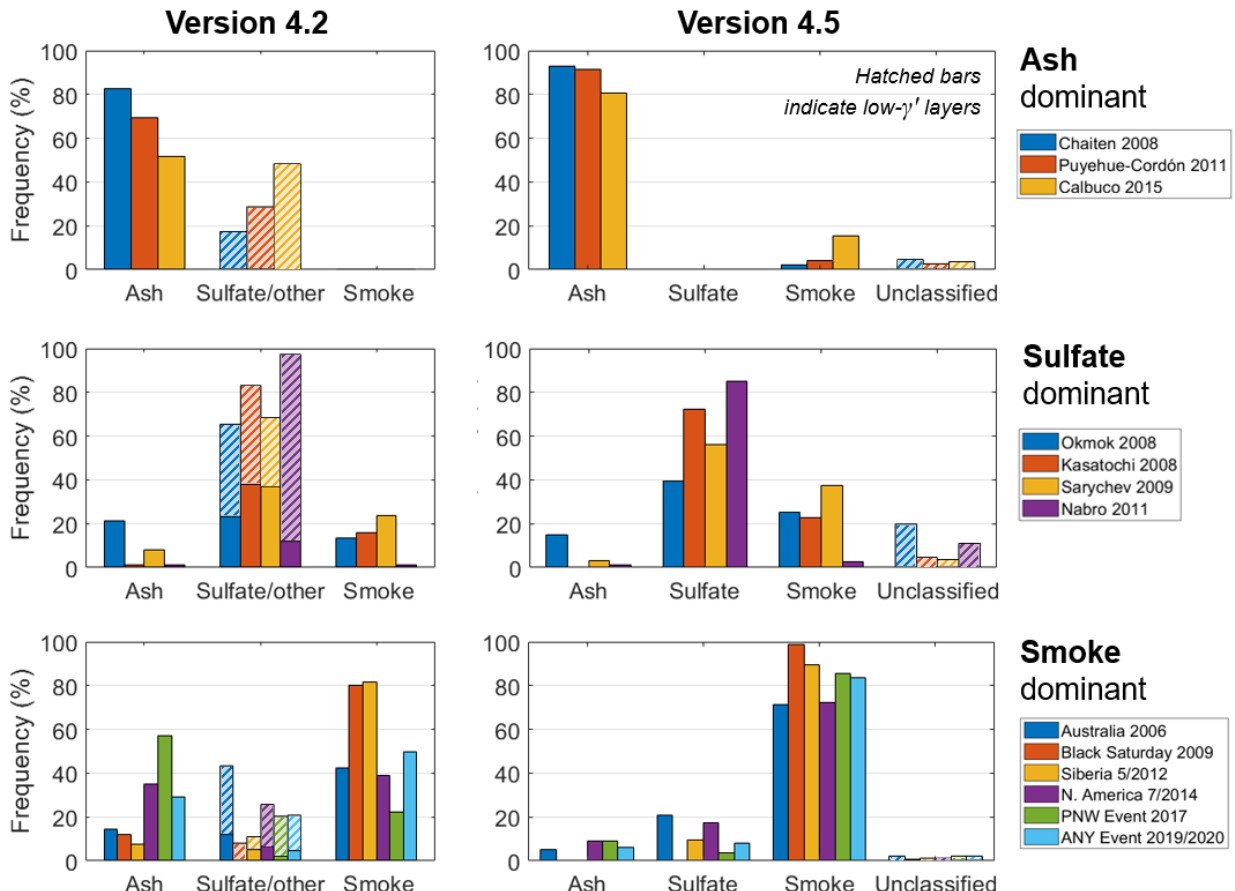

**Figure 11.** Stratospheric aerosol subtype classification frequency for events dominated by ash, sulfate, and smoke for V4.2 (left) and V4.5
(right) based on manually identified layers in Table 1. Low-γ′ layers indicated by hatched bars.

## 5 Performance assessment

We now take a closer look at the geographical and time-evolution of classifications for specific stratospheric aerosol events
to assess the performance of the algorithm. Whereas the previous section summarized classifications for manually identified
volcanic ash, sulfate, and smoke layers, we now evaluate the classifications for all aerosol layers detected in the stratosphere
following major aerosol injections. As before, we have selected events where the dominant aerosol subtype is known based



on literature sources. Broadening our evaluation to include all aerosol layers detected in the stratosphere rather than manually identified layers gives a sense of the fidelity of the algorithm in the wide range of scenes that CALIPSO encounters. Subtype classification frequencies and depolarization ratio statistics are for night and day layer detections

collectively unless otherwise noted.

**5.1 Ash dominated events**

The stratospheric aerosol typing algorithm performs exceptionally well at identifying volcanic ash. One event dominates the CALIPSO record for this aerosol type: the Puyehue-Cordón Caulle eruption on 4 June 2011. Located in southern Chile (40.6° S, 72.1° W), the volcano injected an estimated ~0.4 Tg of ash into the atmosphere of the southern hemisphere

(Bignami et al., 2014) to altitudes of 12-14 km (Ulke et al., 2016). The plume circumnavigated the globe, affecting air traffic in multiple countries (Wunderman 2012). A strong signature of ash was evident based on elevated CALIOP depolarization ratios and ash retrievals by MODIS and IASI (Klüser et al., 2013; Vernier et al., 2013; Bignami et al., 2014; Maes et al., 2016; Prata et al., 2017, 2020; Christian et al., 2020).

         CALIOP detected ash primarily after 15 June 2011. (The CALIPSO payload was turned off due to adverse space

weather during 6–15 June). Stratospheric aerosol layer detections in Fig. 12a show that the layers detected during 15–28 June 2011 span all longitudes, primarily south of 30° S. These layers were mainly detected from 8 to 14 km (Fig. S1a). The median $\delta_p^{est}$ is 0.34 for all stratospheric aerosol layers in the southern hemisphere (Fig. 13), excluding those given the PSA subtype. Therefore, ash is the dominant subtype assigned at a frequency of 84.3 %, with smoke and sulfate classifications at rates of 3.3 % and 0.2 %, respectively. Smoke misclassifications are expected as well, given the overlap in distributions of

$\delta_p^{est}$ between the volcanic ash and depolarizing smoke regimes. In addition, the daily median $\delta_p^{est}$ of detected layers steadily decreased from 0.35 to 0.25 during days +15 to +45 after 4 June (−0.02 /week), possibly contributing to the number of smoke misclassifications. Nonetheless, ash remained the dominant aerosol subtype classification for over 45 days past the eruption (Fig. 12b).



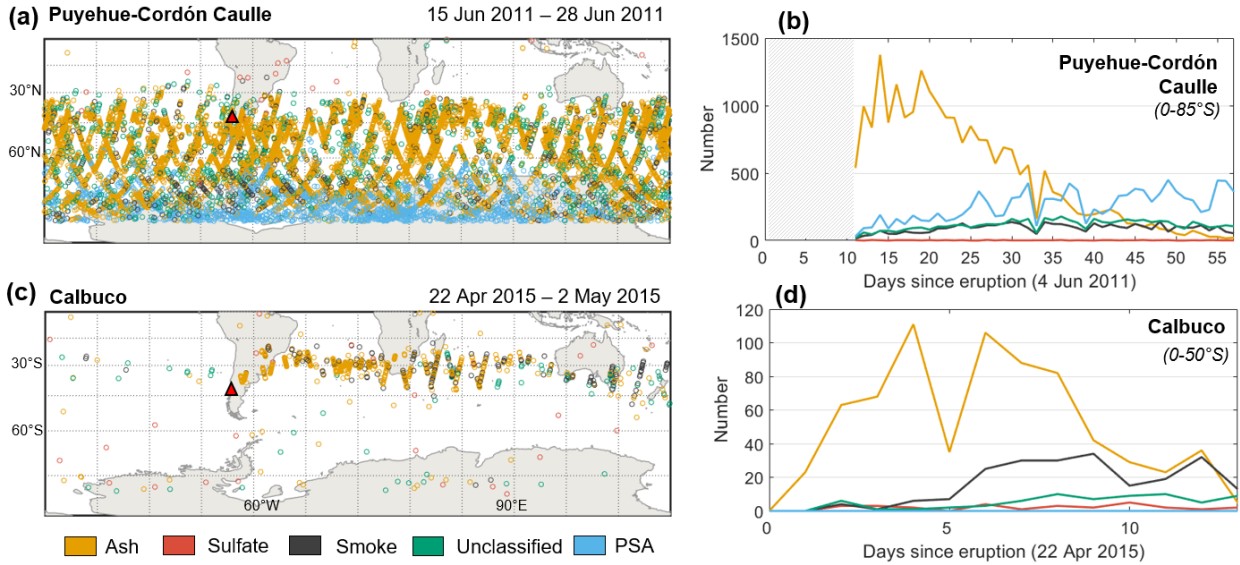


**Figure 12.** For ash-dominated events; (left) locations of subtype classifications following first CALIOP detection and (right) time-history summation of subtype classifications. Volcano locations denoted by red triangles. Hatched areas indicate missing CALIOP data. The payload was down for testing and spacecraft maneuvers during part of 27 April 2015, causing the reduced number of layer detections on that date in panel (d).


The time period of the Puyehue-Cordón Caulle eruption coincided with the beginning of PSC season over Antarctica and many PSA classifications are evident in Fig. 12a. During the first two weeks after the main eruption, the volcanic ash and PSA classifications were mostly separated in altitude. Figure 14 compares the layer top altitudes of ash classifications south and north of 50° S to that of PSA classifications which can only be south of 50° S. Ash layers are

confined below 14 km whereas 64 % of the PSA classifications are above this altitude. These higher altitude layers are likely legitimate PSA classifications due to their low temperatures (−70 °C) and low depolarization ratios (median ~0.02), consistent with liquid supercooled ternary solution droplets (Pitts et al., 2011). The accuracy is questionable for the remaining 36 % of PSA classifications below 14 km, accounting for 5 % of all layers detected at these altitudes. These layers have a median depolarization of 0.32, consistent with ash, though some mixtures of PSC particles also have elevated

depolarization ratios (Pitts et al., 2011). Given the coincident altitude with the ash plume, some of these layers are likely to be ash misclassified as PSA.



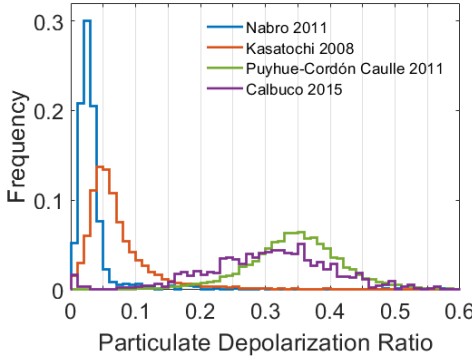

**Figure 13.** Estimated particulate depolarization ratio for all unique stratospheric aerosol layers detected in the first two weeks following first CALIOP observation of the volcanic plume. PSA and layers with |CAD| < 20 are excluded.

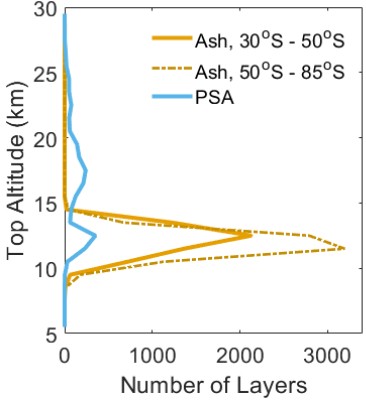

**Figure 14.** Top altitudes for layers classified as ash and PSA during 15–28 June 2011.

The second major volcanic ash event we evaluate is the Mount Calbuco eruption on 22–23 April 2015. Also a Chilean volcano, Mount Calbuco (41.3° S, 72.6° W) injected an estimated 3 Tg of volcanic ash (Marzano et al., 2018) and 0.2–0.4 Tg of $SO_2$ with initial plume heights reaching 18–21 km (Pardini et al., 2018; Zhu et al., 2018). Lidar and AERONET observations from São Paulo acquired just days after the eruption indicated the presence of both sulfate and ash (Lopes et al., 2019), though the bulk of sulfate formation did not complete until the second half of May (Bègue et al., 2017). During the first two weeks, however, the presence of volcanic ash in the plume was confirmed by elevated lidar depolarization ratios (Klekociuk et al., 2020) and by negative $10.06 - 12.05$ μm brightness temperature differences measured by the Imaging Infrared Radiometer, also on board the CALIPSO platform (Fig. S2).

CALIOP detections of stratospheric aerosol layers during this time are primarily along 30° ± 10° S from Chile to the western coast of Australia (Fig. 12c). Most of these layers were detected between 12 and 22 km (Fig. S1b). The median



$\delta_p^{est}$ for stratospheric aerosol layers from Calbuco was slightly lower than Puyehue-Cordón Caulle, at 0.31 for the first two

weeks (Fig. 13). These layers experienced a more rapid decline in depolarization of −0.10 /week. This decline was also

observed by the Cloud-Aerosol Transport System (CATS) lidar at 1064 nm (Christian et al., 2020). Ash classifications are

the dominant CALIOP subtype identified during this event, at 69.4 % (Fig. 12d). Sulfate classifications accounted for 2.7 %

of layers detected during these two weeks, which is reasonable based on AERONET observations consistent with sulfate

over Chile (Lopes et al., 2019). Due in part to the broader distribution of $\delta_p^{est}$ compared to Puyehue-Cordón Caulle, smoke

misclassifications occur at a higher frequency of 21.1 % beginning ~5 days following the eruption. Notably, while the

median $\delta_p^{est}$ decreased in days +1 to +5 following the eruption compared to days +6 to +10, the breadth of the $\delta_p^{est}$

distribution remained roughly the same (median absolute deviation (MAD) of 0.057 and 0.053, respectively). It is possible

that these smoke misclassifications are mixtures of ash and sulfate in the same air mass measured by CALIOP, resulting in

intermediate $\delta_p^{est}$ values between the two aerosol types. This cannot be definitively established with CALIOP measurements

alone, however.

**5.2 Sulfate dominated events**

Two major stratospheric sulfate events are selected for assessment. The first is the Nabro stratovolcano in Eritrea (13.37° N,

41.7° E) which erupted on 12–13 June 2011, injecting an estimated 1.5 Tg of $SO_2$ into the upper troposphere-lower

stratosphere (Clarisse et al., 2014). A second injection into the stratosphere on 16 June was inferred based on geostationary

and limb-profiling satellite data (Fromm et al. 2014). The $SO_2$ plume initially traveled east and then followed the Asian

summer monsoon anticyclonic circulation over northern Africa, the Middle East, and Asia for the first two weeks (Fairlie et

al., 2014). Sulfate aerosol then transported to the rest of the northern hemisphere over July and August.

       We focus on the region shown in Fig. 15a. CALIOP detected the majority of the stratospheric plume between 13 to

19 km during the two weeks following the initial eruption (Fig. S3a). The median $\delta_p^{est}$ for these layers is the smallest of all

volcanic events evaluated, at 0.021 (Fig. 13), consistent with various ground based lidars that also measured small

depolarization from the Nabro plume, (Zhuang and Yi, 2016; Noh et al., 2017). This is indicative of sulfate aerosol. Due to

the low $\delta_p^{est}$, CALIOP classifies 82.6 % of these layers as sulfate. Smoke and ash classifications account for 5.4 % and 1.7

%, respectively, consistent with observations suggesting only a small ash component to the eruption (Clarisse et al. 2014).




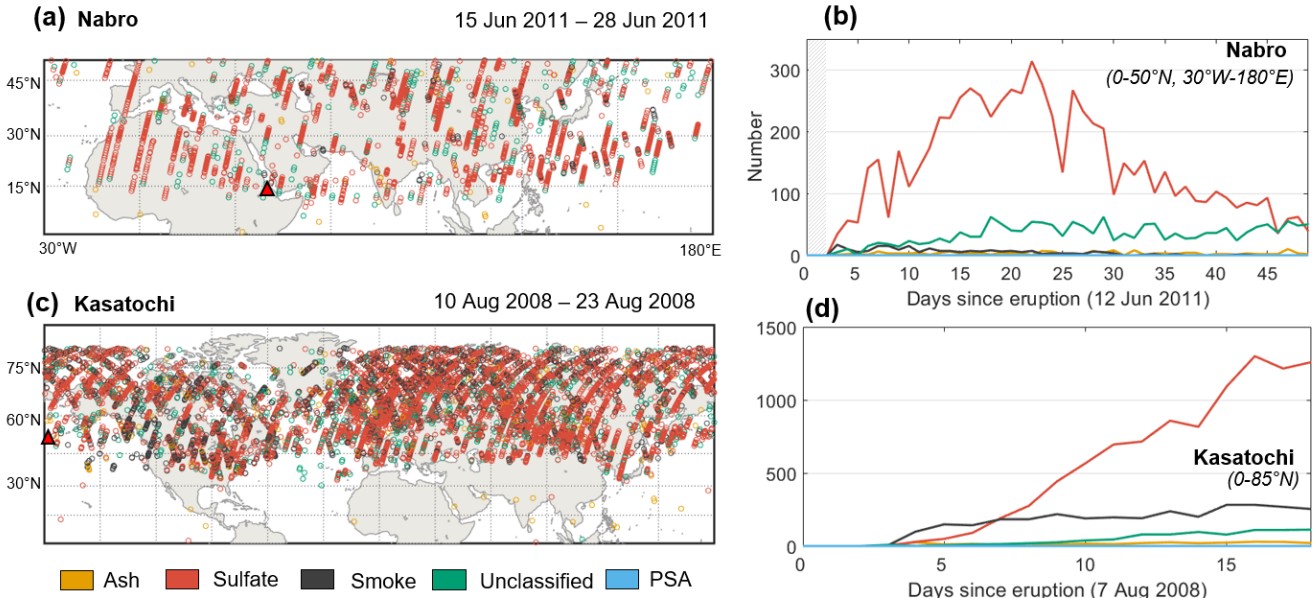

**Figure 15.** Same as Fig. 12, but for sulfate-dominated events.

The second sulfate-dominated event we evaluate had a slightly larger ash component. Kasatochi, an island volcano
along the Aleutian arc (52.17° N, 175.51° W), erupted on 7–9 August 2008, injecting SO₂ and ash up to ~15 km in altitude
(Waythomas et al. 2010). In subsequent days, signatures of volcanic ash and SO₂ were observed spreading eastward over the
Pacific by MODIS, AVHRR, AIRS, and OMI (Corradini et al., 2010; Krotkov et al. 2010). An analysis of AIRS
measurements by Prata et al. (2010) suggests that the total ash mass injected was approximately 25 % smaller than the mass
of SO₂. Based on passive imager retrievals and modeling analyses, the greatest fraction of this ash is believed to have settled
out of the plume during the first week following the eruption (Martinsson et al., 2009; Guffanti et al., 2010; Langmann et al.,
2010). The most long-lasting component of the emission was SO₂ and subsequent sulfate aerosol that persisted for over two
months. At the time, it was the largest injection of SO₂ into the atmosphere in over 17 years, with SO₂ mass estimates of 1.2–
1.7 Tg (Kristiansen et al. 2010; Prata et al. 2010).

The majority of stratospheric aerosol layers detected by CALIOP in the two weeks following its first detection of
the plume on 10 August were above 30° N (Fig. 15c), at altitudes of 9–14 km (Fig. S3b). The median $\delta_p^{est}$ of layers in this
altitude range is around 0.052, leading to a sulfate classification rate of 67.6 %. The smoke and ash classification rates are
24.4 % and 5.8 %, respectively. A secondary peak with a smaller number of layers around 16 to 18 km was also detected
(Fig. S3b). These layers have a somewhat higher median $\delta_p^{est}$ of ~0.062, yielding sulfate, smoke, and ash classification rates
of 52.3 %, 37.8 %, and 3.3 % respectively. Figure 16 shows two examples of these higher altitude plumes over the eastern
Pacific on 14 and 15 August, having median $\delta_p^{est}$ values of 0.161 and 0.109. Due to these elevated depolarization ratios, the



dominant classification was smoke, but it is likely that these are mixtures of ash and sulfate, with the ash component larger in the 14 August observation. All told, the time series of stratospheric aerosol classification shows an appearance of smoke classifications coincident in time with the appearance of sulfate classifications (Fig. 15d). The altitudes of layers given these classifications are also the same (Fig. S3b), suggesting these plumes are from the same event. Taken together with broader

$\delta_p^{est}$ distribution compared to the 2011 Nabro eruption (Fig. 13) these smoke classifications could indicate mixtures of ash and sulfate.

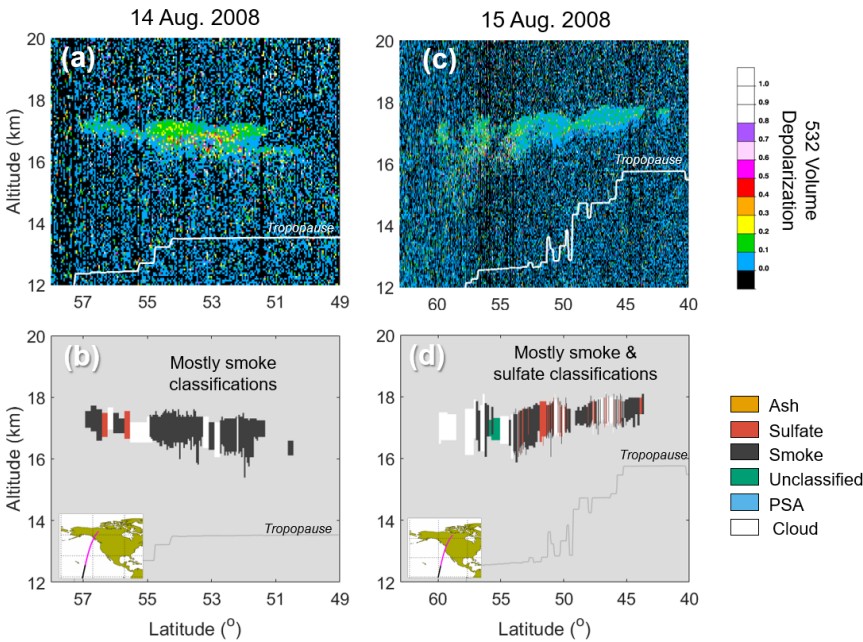

**Figure 16.** Possible ash-sulfate mixtures from the 2008 Kasatochi eruption; 532 nm volume depolarization ratio and V4.5 aerosol type
classifications from the level 2 aerosol profile product on (a, b) 14 August 2008 at ~11:30 UTC, and (c, d) 15 August 2008 at ~10:30 UTC. Inset maps show CALIOP ground track.

**5.3 Smoke dominated events**

In recent years, two major wildfire events demonstrated the massive influence pyroCb activity can have on stratospheric aerosol loading. On 12 August 2017, a series of pyroCbs occurred in northern Washington state, United States and British
Columbia, Canada. Dubbed the "Pacific Northwest (PNW) event", Peterson et al. (2018) estimated that 0.1–0.3 Tg of aerosol mass was injected into the stratosphere on this day. CALIPSO initially measured the plume at 12–14 km on 14 August over northeastern Canada. Several lidar systems over Europe detected the plume over Europe by as early as 10 days later at altitudes spanning 15 to 20 km (Ansmann et al., 2018; Haarig et al., 2018; Khaykin et al., 2018; Hu et al., 2019). Analysis of CALIPSO observations by Khaykin et al. (2018) show the smoke plume had circumnavigated the globe by 30
August, affecting the stratosphere in the entire northern hemisphere above 30° N.





During the first two weeks of the PNW event, the CALIOP median $\delta_p^{est}$ was 0.157 for all stratospheric aerosol layers detected in the northern hemisphere. Consequently, for night & day, 76.6 % were classified as smoke, with 9.1 % and 10.4 % misclassified as sulfate and ash, respectively. During this timeframe, these stratospheric smoke layers were primarily detected over northeast Canada, across the north Atlantic, and well into northern Asia (Fig. 17a) at altitudes of 9 to 19 km

(Fig. S4a). The time series of smoke layers classification shows a maximum 10–15 days after the initial event (Fig. 17b). Ash misclassifications primarily occur in daytime orbits where additional solar noise broadens the variability of $\delta_p^{est}$, more than doubling the MAD (Fig. 18a). As a result, 22.1 % ash misclassifications occur at day compared to 1.8 % at night. Smoke classifications fare much better at night, at a frequency of 90.1 %.

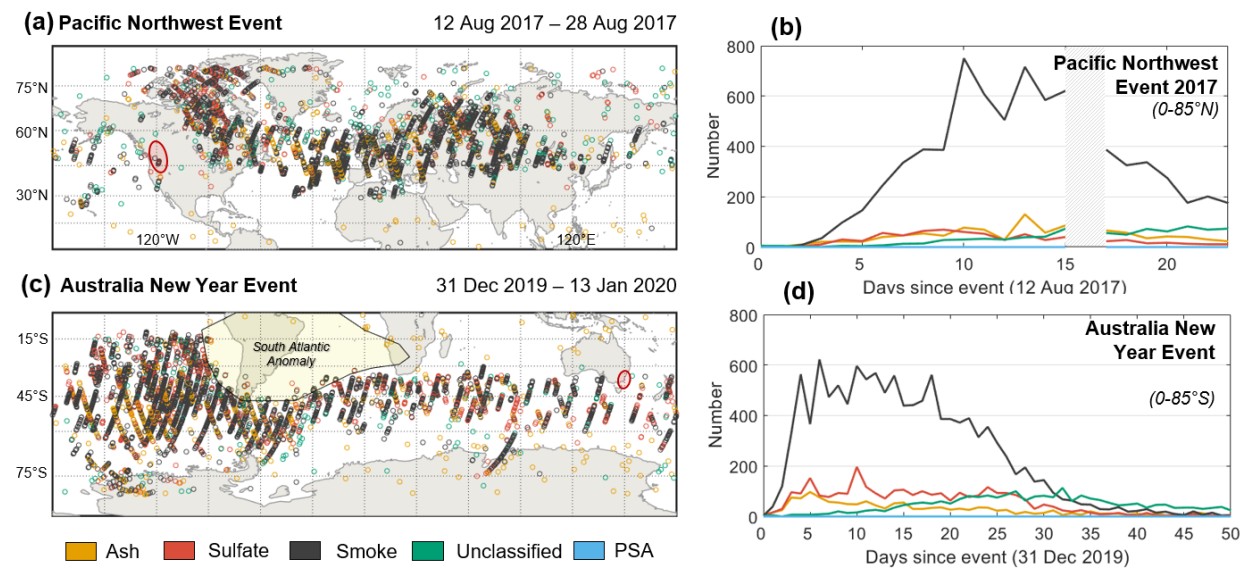

**Figure 17.** Same as Fig. 12, but for smoke-dominated events. PyroCb locations denoted by red ovals. Approximate area affected by SAA denoted by yellow polygon.

A second major wildfire event occurred just 2.5 years later in southeastern Australia. From 29 December 2019–4

January 2020, a series of massive pyroCbs injected smoke as high as 16 km (Kablick et al., 2020). Dubbed the 2019/2020 Australian New Year (ANY) event, preliminary estimates of the injected aerosol mass are even larger than the PNW event, ranging from 0.2–0.9 Tg (Peterson et al., 2019; Khaykin et al., 2020). The smoke plumes primarily traveled eastward during the first month, ultimately ascending to heights of over 30 km in February as the smoke absorbed solar radiation, heating the surrounding air and affecting atmospheric dynamics locally (Allen et al., 2020; Kablick et al., 2020). Depolarization of the

pyroCb plume from the ANY event was also elevated relative to tropospheric smoke. A Raman lidar in Punta Arenas, Chile measured 532 nm depolarization ratios of 0.14–0.22 during January 2020 (Ohneiser et al., 2020) with indications of an increase in depolarization with time (Christian et al., 2020).


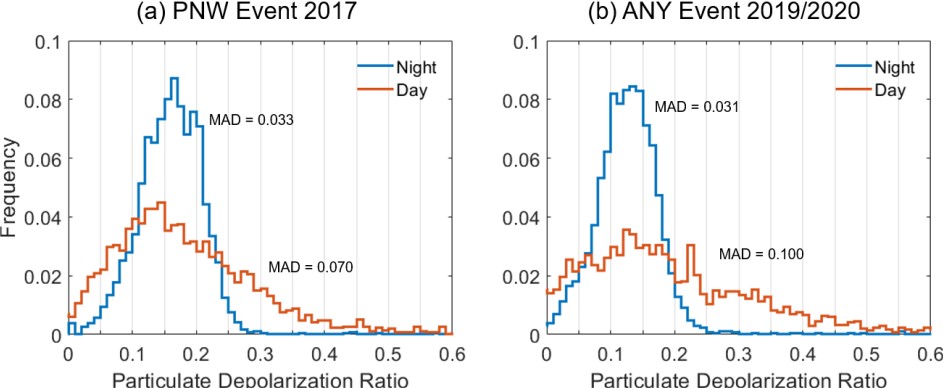

**Figure 18.** Night and daytime estimated particulate depolarization ratio distributions for all stratospheric aerosol layers detected during the first two weeks after (a) the PNW event (northern hemisphere) and (b) the ANY event (southern hemisphere). Low-$\gamma'$, PSA, and layers with |CAD| < 20 are excluded.

During the first two weeks of January 2020, CALIOP detected stratospheric smoke layers primarily over the southern Pacific Ocean yet spanning all longitudes (Fig. 17c) at altitudes of 11 to 22 km (Fig. S4b). Median $\delta_p^{est}$ was 0.125 for stratospheric aerosol layers in the southern hemisphere, somewhat smaller than the Raman lidar measurements in Chile. However, the distributions of $\delta_p^{est}$ are quite broad, in particular during daytime (Fig. 18b). Owing to the elevated values of $\delta_p^{est}$, the night & day stratospheric smoke classification frequency was 73.1 %, with misclassification frequencies of 15.5 % and 9.2 % for sulfate and ash, respectively. As with the PNW event, the daytime $\delta_p^{est}$ distribution was broader compared to night with a strong skew toward larger values. Consequently, ash misclassifications for the ANY event are more frequent in the daytime, at a rate of 26.9 % compared to 0.6 % at night. The most influential factor driving the broader daytime $\delta_p^{est}$ distribution is sunlight reflecting from high albedo targets at lower altitudes such as stratocumulus in the PBL and snow-covered surfaces (e.g., Antarctica in the ANY event). This reflected sunlight enhances noise throughout the profile overhead, thereby increasing the variability of depolarization ratio measurements. As a final comment regarding the $\delta_p^{est}$ distribution, we turn our attention to the nighttime distributions in Fig. 18, which more closely resemble natural variability. Because aging is known to affect smoke depolarization in the troposphere (Burton et al., 2015), the breadth of these distributions could suggest that, in part, a diversity in aging was sampled in these hemispheric averages. This is speculative however, since a full understanding of the composition and evolution of aerosol within pyroCb plumes is an active area of research.

Returning to the geographic distribution of smoke from the ANY event, the map in Fig. 17c shows during the first two weeks, most smoke layers are detected over the southern Pacific Ocean as far north as the equator, though most are detected south of 30° S at all longitudes. Note that the majority of layers detected in the SAA are excluded from our analysis



by the minimum laser energy requirement (Sect. 2) we impose to avoid the detrimental influence of low laser energy shots that are prevalent in this region since mid-2016 (CALIPSO Data Advisory Page, 2018). The maximum number of smoke classifications occurs during the first 4 weeks, with an increase in unclassified low-$\gamma'$ layers (Fig. 17d). These low-$\gamma'$ layers

become the dominant classification in early February 2020. However, the depolarization ratios remained notable. Figure 19 shows the coherent "bubble" of smoke southwest of the southern tip of Chile on 31 January that was examined by Allen et al. (2020) and Khaykin et al. (2020). They found this bubble rose to 35 km in subsequent months due to dynamics associated with absorption of solar radiation. The CALIOP CAD algorithm struggled with this scene, classifying much of the feature as ice cloud due to the elevated depolarization and spread in moderate color ratio values (median ± MAD of 0.44 ± 0.23). The

CAD probability density functions have a fair amount of overlap in the color ratio dimension for features with elevated depolarization ratios at high altitudes (Liu et al., 2018), thereby contributing to these CAD errors. However, most of the layers classified as aerosol in V4.5 are correctly given the smoke subtype (65 %), whereas ash was the dominant subtype given in V4.2, with 8 % smoke.

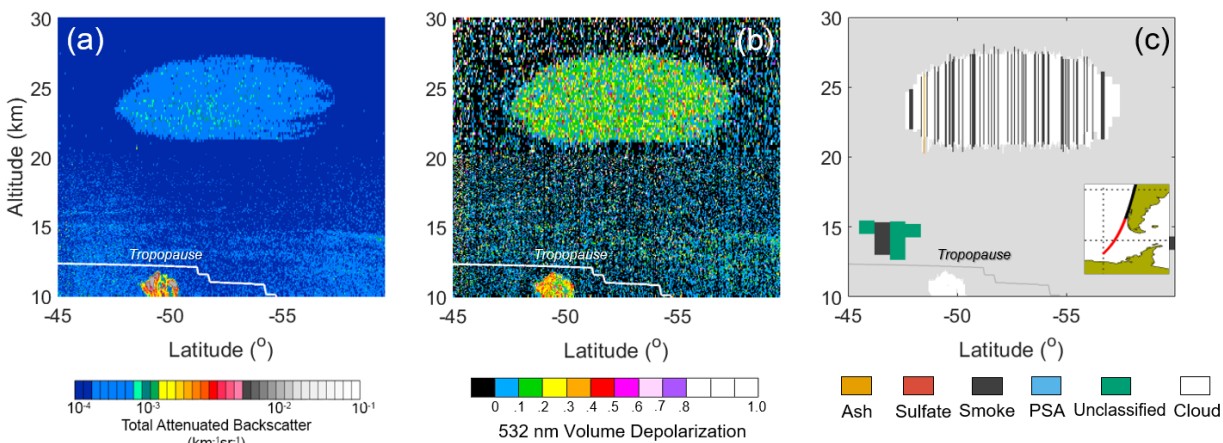


**Figure 19.** PyroCb smoke plume from ANY event, 31 January 2020 at ~6:30 UTC: (a) 532 nm total attenuated backscatter, (b) 532 nm volume depolarization ratio, and (c) V4.5 aerosol subtype and cloud classification from the level 2 aerosol profile product. Inset map shows CALIOP ground track.

**5.4 Unclassified layers and false-positive feature detections**

For features identified as stratospheric aerosols by the CALIOP CAD algorithm, the frequency of unclassified layers is bound by the low-$\gamma'$ threshold at the high end and by the feature detection sensitivity at the low end. Regardless of the actual aerosol type, all stratospheric aerosol layers can be assigned this classification because eventually, due to sedimentation and diffusion, their concentrations in the atmosphere will decline until they are no longer detectable by the CALIOP feature finder. By design, the lowest-quartile $\gamma'_{532}$ metric causes 25 % of stratospheric aerosol layers in the CALIOP data record to

be unclassified on average (Sect. 4.4). Though the number of unclassified stratospheric aerosols peaks legitimately during





major events, there is a "background" number reported during quiescent periods associated with false layer detections (Fig. 20). As a first-order estimate, during the year 2013 when there were no major stratospheric aerosol injections, suspected false-positive feature detections above 20 km occurred in 0.4 % of profiles at night and 1 % at day, globally. The location for these layers primarily occurs inside the SAA and is randomly distributed at a lesser frequency over the globe (Fig. S5).
Because they do not occur within specific latitude bands, as would legitimate layers associated with specific events, they are likely due to false layer detections caused by radiation spikes and background noise. The aerosol subtypes of these false layer detections at night are 54 % ash, 30 % sulfate, 5 % smoke, and 11 % unclassified. During the day, these frequencies change to 84 % ash, <1 % sulfate, 3 % smoke, and 13 % unclassified. False layer detections above 20 km and have characteristically low CAD scores (97 % have |CAD| < 20 indicating no confidence in cloud-aerosol discrimination
accuracy). They can be readily rejected using the CAD score as we have done throughout this paper.

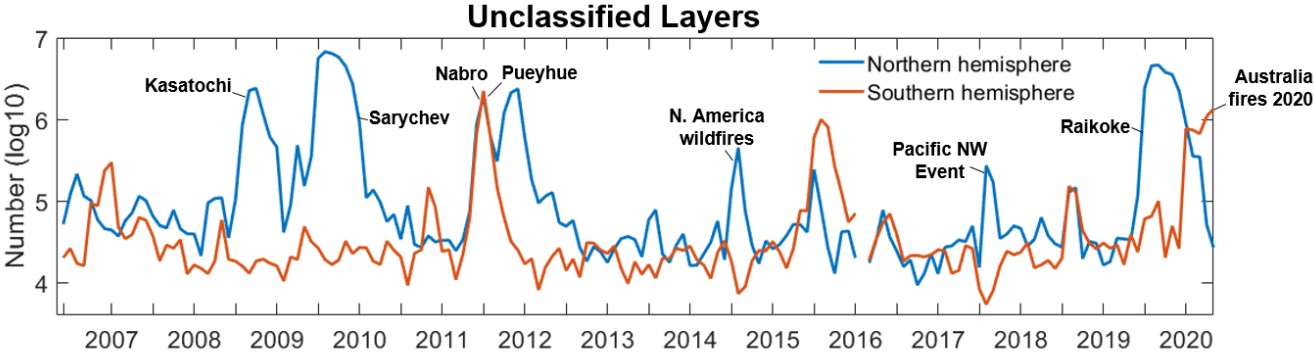

**Figure 20.** Number of unclassified layers in the northern (blue) and southern (red) hemisphere, excluding the SAA and layers with |CAD score| < 20. Computed using V4.2 integrated attenuated backscatter with V4.5 low-$\gamma'$ thresholds, so the figure serves as a close
approximation to V4.5.

## 6. Conclusion

The stratospheric aerosol subtyping algorithm has been updated for the V4.5 release of the CALIOP level 2 data products. Following the previous V4.2 release, it became clear that several aspects of the newly introduced stratospheric aerosol subtyping algorithm needed further refinement. This paper describes the changes the CALIPSO project has implemented to
improve aerosol subtyping in the stratosphere and characterized the performance of the refined algorithm based on well-documented events. The changes include: removing the use of integrated attenuated backscatter color ratio, so that the algorithm now discriminates between volcanic ash, sulfate, and smoke solely based on depolarization; increasing the depolarization threshold to discriminate between volcanic ash and sulfate; separating the V4.2 sulfate/other subtype into sulfate and unclassified subtypes; lowering the low-$\gamma'$ threshold for identifying weakly scattering, unclassified, layers; and
increasing the 532 nm lidar ratio for volcanic ash to a value consistent with the current state of knowledge. As a





consequence, these changes improve the discrimination capability between volcanic ash and smoke by better accounting for the depolarizing nature of smoke often observed for layers associated with pyroCb activity. Sulfate classifications now solely identify layers having low depolarization ratios, a characteristic of sulfate aerosol. Our analysis also postulates that volcanic layers classified as smoke can indicate mixtures of sulfate and ash. Finally, weakly backscattering features have been

relegated to a new "unclassified" subtype for which the signal-to-noise ratio is considered insufficient to reliably discern the true type.

        The performance of the revised algorithm is very good for volcanic ash layers, with 84 % correctly classified during the ash-dominated Puyehue-Cordón Caulle eruption of 2011. This is no surprise, given the strongly depolarizing nature of volcanic ash. Sulfate classifications are also dominant for events having a strong sulfate component. However, the

interpretation of the CALIOP stratospheric aerosol classification requires some extra care for sulfate-dominated scenes with some ash component. Sulfate/ash mixtures are misclassified as smoke for nearly one-third of these layers and there exists the possibility for legitimate smoke layers to be misclassified as sulfate due to the overlap in the depolarization ratio distributions for these two types (a combination of natural variability and measurement noise). Smoke classification performance for events dominated by pyroCb activity was also very good, with most layers classified as smoke. There

remains a moderate number of smoke layers that are misclassified as sulfate and ash. In particular, misclassification frequencies of smoke as volcanic ash are substantially higher at day than at night (~27 % vs. 1 %, respectively), due to reflected sunlight from lower altitude high-albedo features that adds substantial noise to the column, broadening the distribution of depolarization ratios. Additionally, smoke transported from the troposphere into the UTLS by self-lofting rather than pyroCb activity will likely be misclassified as sulfate due to their similarly low values of depolarization.

Researchers should be aware of these potential artifacts when performing automated analyses with CALIOP V4.5 level 2 data.

        It is important to recognize that although the aerosol subtyping algorithm performs very well for ash, sulfate, and depolarizing smoke in the stratosphere, aerosol subtyping is less satisfactory for these same aerosol types below the tropopause largely because no attempt is made to identify them in the troposphere. There, volcanic ash will inexorably be

misclassified as dust, depolarizing smoke mostly misclassified as polluted dust, and volcanic sulfate misclassified as elevated smoke (Kim et. al, 2018). These misclassifications occur because it is difficult to discriminate among these aerosol types in a robust automated manner given the limited number of CALIOP observables. The critical information for the CALIOP stratospheric aerosol subtyping algorithm is the high altitude of the tropopause, which most often rules out the possibility of all but a few subtypes. More sophisticated instrumentation will improve discrimination capability in the

troposphere, such as high spectral resolution lidar with depolarization sensitivity at 355 nm, 532 nm, and 1064 nm (e.g., as in Burton et al., 2015) or combined lidar plus passive instrument retrievals. Additionally, combining $SO_2$ and CO measurements from other sensors could help differentiate between ash/sulfate mixtures and smoke. Geostationary radiances could help differentiate between smoke and volcanic aerosol using the split window technique. Given the aviation hazards posed by volcanic ash and the climate implications of sulfate and stratospheric smoke injections, space-based lidar retrievals



stand to provide valuable vertically resolved information to disaster response agencies and climate modelers. Our hope is that this work provides a meaningful steppingstone toward more sophisticated solutions in future missions.

**Data availability**

CALIOP data are available through the NASA Langley Research Center Atmospheric Science Data Center, https://asdc.larc.nasa.gov/ (last access: 7 January 2022): V4.1 level 1B (NASA/LARC/SD/ASDC 2016b); V4.2 level 2
aerosol layer, aerosol profile, and vertical feature mask products (NASA/LARC/SD/ASDC 2018a, 2018b, 2018c); V4.5 level 1B (NASA/LARC/SD/ASDC 2022a), V4.5 level 2 aerosol layer, aerosol profile, and vertical feature mask products (NASA/LARC/SD/ASDC 2022b, 2022c, 2022d).

**Author contributions**

The CALIPSO stratospheric aerosol subtyping algorithm described in this manuscript was conceived by JT, MV, JK, AO,
and DW, with contributions from the CALIPSO mission team, JV, MP, and Lamont Poole. Formal analysis and visualizations were prepared by JT, except for Sect. 4.5 which was prepared by JK. Writing was led by JT with extensive contributions by MV, JK, MK, and JV. All V4.5 pre-release data was generated by BG. CALIPSO V4.5 level 2 software modifications for the stratospheric aerosol subtyping algorithm were performed by BM with support from the CALIPSO Data Management Team.

**Acknowledgements**

Volcano geographic locations acquired from Smithsonian Institution National Museum of Natural History Global Volcanism Program https://volcano.si.edu/.

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
