# Peer review of "The CALIPSO version 4.5 stratospheric aerosol subtyping algorithm"

_Atmospheric Measurement Techniques, 2022_

## Author Comment (AC1)

**Response to Anonymous Referee #1 Comments**

Review of The CALIPSO version 4.5 stratospheric aerosol subtyping algorithm by Tackett et al.

This description of the changes to the CALIPSO version 4.5 stratospheric aerosol subtyping algorithm is well written and easy to follow with an appropriate amount of detail, and the algorithm changes it describes appear to be logical and useful upgrades to the algorithm. I especially appreciated that almost every time I had a question while reading, the answer came within half a page of my question, which I think is a sign of good flow. The manuscript is particularly strong on explaining the logic of the changes to the classification algorithm. I have only two significant critiques.

First, there isn't much analysis or discussion nor validation of the new lidar ratio, which, along with changes to the classification methodology, was also included as one of the algorithm changes highlighted in this manuscript. I realize that it is nearly impossible to validate the lidar ratios selected for the CALIPSO algorithm, since any available independent information is probably already included in the decision on what range of values to set it at. However, I think some more discussion of the impact and of the sensitivity would be appropriate.

Second, I think there should probably be some emphasis on the idea that the errors in the estimated particulate depolarization ratio are dominated by systematic rather than random errors, and possibly some analysis or demonstration of why estimated particulate depolarization ratio is nevertheless necessary for classification or better than volume depolarization, which is more likely to be strongly dominated by random rather than systematic errors.

Thank you for your positive comments and thorough review. Addressing these questions has improved the clarity of our manuscript. We appreciate your feedback.

It is certainly a good idea to expand upon how aerosol type selection and lidar ratio selection errors propagate into lower layers to bolster our argument to choose as accurate values as possible. We added a new paragraph to Sect. 4.5 that explains the types of errors that would arise from incorrect lidar ratio selection.

Indeed, there are systematic uncertainties in CALIOP's estimated particulate depolarization ratios, and we agree that they should be discussed in the manuscript. However, the relative contribution of systematic errors compared to random errors in CALIOP data are not the same as for the HSRL error budget discussed in Burton et al., 2015. The equation below, adapted from Eqn. A5 of Burton et al. 2015, shows the error propagation for Eqn. 2 in our manuscript.

$$\left(\frac{\Delta\delta_p^{est}}{\delta_p^{est}}\right)^2 = F_R\left(\frac{\Delta R'}{R'}\right)^2 + F_{\delta_v}\left(\frac{\Delta\delta_v}{\delta_v}\right)^2 + F_{\delta_m}\left(\frac{\Delta\delta_m}{\delta_m}\right)^2 \tag{R1}$$

In our formulation, $R'$ is the mean attenuated scattering ratio computed between layer top and base, whereas R in the Burton paper is a range-resolved unattenuated scattering ratio; i.e., $R'(r) = R(r) \times T_p^2(r)$, where $T_p^2(r)$ is the signal attenuation due to any particulates (clouds and aerosols) in the

optical path between the lidar and measurement range r. Burton et al. (2015) can use R because the HSRL directly measures R. CALIOP directly measures R′, but can only obtain R after making in-layer attenuation corrections (i.e., computing $T_p^2(r)$), which requires a priori specification of a lidar ratio. These lidar ratios are one of the primary outputs of the stratospheric aerosol subtyping scheme. The calculation of $\delta_p^{est}$, on the other hand, is one of the essential inputs. Additional details have been added to Sect. 3 describing the relationship between R and R′.

Here, we will expand the attenuated scattering ratio relative uncertainty term to include both random and systematic errors. Equation R2 below shows the sum of errors for the layer attenuated scattering ratio for CALIOP.

$$\left(\frac{\Delta R'}{R'}\right)^2 = \left(\frac{\Delta \beta'}{\beta'}\right)^2 + \left(\frac{\Delta C}{C}\right)^2 + \left(\frac{\Delta \beta_m}{\beta_m}\right)^2 + \left(\frac{\Delta T_m^2}{T_m^2}\right)^2 + \left(\frac{\Delta T_{O_3}^2}{T_{O_3}^2}\right)^2 \tag{R2}$$

The first term on the right-hand side (RHS) represents the random error in attenuated backscatter based on the shot noise in the CALIOP detectors, including noise in the background signal (Liu et al., 2006). Because the CALIOP level 2 scene classification algorithms evaluate layer-averaged or layer-integrated quantities, the relevant estimate of random error is based on the aggregate of data within the layer boundaries. The level 2 products report the integrated attenuated backscatter ($\gamma'$) uncertainty for each layer, based on the formulation Liu et al., 2006 (see also https://www-calipso.larc.nasa.gov/resources/calipso_users_guide/tools/idl/AttenBksUncertainties.pdf). From this, the random *relative* error in the layer-average attenuated backscatter $\Delta \beta'/\beta'$ is identical to the layer-integrated attenuated backscatter, $\Delta \gamma'/\gamma'$.

The distribution of $\Delta \gamma'/\gamma'$ is shown below for all nighttime stratospheric aerosol layers evaluated in the manuscript, segregated by the horizontal average required for layer detection. Features classified as PSA or with |CAD| < 20 are excluded. In the best SNR situation (5 km feature detection at night), the random error is greater than 10% for 90% of the layers detected at that resolution.

[Figure]

The remaining terms on the RHS of Eqn. (R2) are the systematic errors in calibration, molecular backscatter, molecular two-way transmittance, and ozone two-way transmittance. Molecular and ozone related backscatter and attenuation terms are derived from data provided by MERRA-2. The relative uncertainties in the 532 nm calibration are estimated as $1.6 \pm 2.4\%$ at night and $1.0 \pm 3\%$ at day (Kar et al., 2018; Getzewich et al., 2018). Relative uncertainty of molecular backscatter is estimated as 1% or less (Campbell et al., 2015). The combined uncertainty in molecular and ozone two-way transmittance is typically 0.5% or less (Powell et al., 2009). When summed in quadrature, the contribution of systematic errors to the scattering ratio uncertainty is around 2% which is less than the random error for the vast majority of stratospheric aerosol layers that the CALIOP level 2 scene classification algorithm needs to evaluate.

We agree however that a better explanation should be given to section 3.1 as to why the estimated particulate depolarization ratio is preferrable to the volume depolarization ratio. As pointed out by the reviewer, using the former introduces more susceptibility to propagated error very low scattering ratios. To address this, we added details of the rationale adopted by the CALIPSO project to section 3, including a discussion of the overestimates expected by using the attenuated scattering ratio versus the unattenuated scattering ratio in the estimated particulate depolarization ratio calculation.

Liu, Z., et al., 2006: Estimating Random Errors Due to Shot Noise in Backscatter Lidar Observations, Appl. Opt., 45, 4437-4447.

Specific comments on content

L58 mentions the algorithmic motivation for getting the right aerosol type is the need to correct the overlying attenuation. How sensitive are the retrievals lower in the atmosphere to having the correct type in the stratosphere? That is, I realize there is a one-to-one relationship between error in layer lidar ratio and the error in the AOT for the same layer, but can the authors describe how much the uncertainty for lower layers is increased given the distribution of stratosphere layer optical thickness? This would address not just correct typing, but also how much impact changing the volcanic ash lidar ratio would be expected to have.

We chose to add this information to Sect. 4.5 which discusses lidar ratios to avoid bogging the introduction down in too many details. Here is the new paragraph that now starts off Sect. 4.5:

"As emphasized in the introduction, the algorithmic motivation for improving aerosol subtyping is to ensure that a representative lidar ratio is selected for that subtype, thereby yielding accurate extinction retrievals. Because CALIOP retrievals operate from top-down, any errors in the retrieved extinction due to incorrect lidar ratio selection will propagate into underlying layers. The previous subsections described the improvements we made to increase the likelihood that the correct stratospheric aerosol subtype (i.e., correct lidar ratio) will be selected. However, it is also important that the lidar ratio for these subtypes are representative of what is observed in nature. A full, detailed accounting of error propagation in CALIOP extinction retrievals is given in Young et al., 2013, including the impacts of incorrect lidar ratio selection. One key take-away from that paper is that the relative error in retrieved AOD equals the relative error in the lidar ratio for layers having low optical depths, typical of aerosol. A second key take-away is that the error in retrieved

AOD due to incorrect lidar ratio selection behaves as a systematic bias in the retrieval of optical depth for underlying layers. The magnitude of the error in AOD for the underlying layer depends on its $R'$, AOD, and the magnitude of the error in lidar ratio of the overlying layer. In general, though, the sign of the AOD bias is the same as the sign of the error in lidar ratio selection of the overlying layer (i.e., selecting too low of a lidar ratio causes an underestimate in AOD of layers at lower altitudes). Feature detection accuracy for lower layers can also be degraded if overlying attenuation is not correctly accounted for. Clearly, accurate lidar ratio selection is critical for elastic backscatter lidar retrievals."

To link this discussion specifically to the change in ash lidar ratio, this sentence is added later in the section describing the new, larger lidar ratio, "This will increase the retrieved AOD for ash layers by ~39 % and prevent underestimates of optical depth for underlying layers."

L179-180 includes the attenuated molecular backscatter, "calculated from the MERRA-2 model". How can the particulate attenuation of the molecular bacskcatter be caculated accurately given there is no measurement of the attenuation? Or is the particulate attenuation left out? I hope this will be made more explicit, including some comment about what errors or uncertainties are likely. It seems logical that if the particulate attenuation is not accounted for, then R' will have a systematic low bias (compared to true aerosol backscatter ratio), and estimated particulate depolarization would have a high bias. Is that correct?

That is correct – particulate attenuation is not accounted for within the layer. This information has been explicitly discussed in a dedicated paragraph in Sect. 3 with the new Fig. 3(a), shown below, that demonstrates the high bias that arises in the estimated particulate depolarization ratio.

[Figure]

L179-180 A more minor point here: the equation and text suggest that the ratio of attenuated backscatter quantities is taken before doing the average. In general, I would think it was better to take the average of each noisy signal and then the ratio, to limit noise magnification. Perhaps this is irrelevant because the denominator is modeled on a coarser scale than the measurements anyway?

We agree that taking averages of noisy signals prior to taking the ratio is essential for limiting noise magnification. You are exactly right…because the molecular attenuated backscatter in the

denominator is a modeled value on a much coarser resolution that CALIOP measurements, it is not necessary to average prior to computing the ratio for this quantity.

Figure 3. I'm curious to see a panel which shows a representative selection of known PSA, which would help illustrate why the PSA selection must be done before thresholding on depolarization.

This suggestion was very helpful. We added the figure below to this section which showed quite a broad range in depolarization values. With this as a basis, we modified our rationale in the text to, "These layers are often detected adjacent to features classified as cloud and have depolarization levels that overlap with the expected ranges for sulfate, smoke, and ash (Fig. 6)… The PSA classification prevents these layers from being misclassified as volcanic ash or sulfate when none exists."

[Figure]

**Figure 6.** Same joint distributions as Fig. 4, but for layers classified as PSA in the southern hemisphere in 2013, based on V4.5 data.

L274 "based on elevated depolarization". A quick look at the Prata et al. reference that's included in the table suggests that there's supporting information from other sensors. If so, consider including some discussion of supporting evidence in the text here.

The text was revised as follows to provide supporting information.

"The new events include two volcanic eruptions: sulfate layers from the July 2009 Sarychev Peak eruption and ash layers from the April 2015 Mount Calbuco eruption. The plume of volcanic aerosol from Sarychev Peak primarily consisted of sulfate based on AIRS retrievals and in-situ aircraft measurements (Prata et al., 2014; Andersson et al., 2013). The Mount Calbuco eruption, discussed in detail in Sect. 4.1, injected large quantities of ash and a lesser amount of $SO_2$ into the atmosphere (Marzano et al., 2018) resulting in sulfate formation about a month after the eruption (Bègue et al., 2017). To sample the ash component in the joint distribution, we only selected layers within the first two weeks of the eruption; the CALIOP depolarization was elevated for these layers, consistent with ash."

L274 "based on elevated depolarization". Given an a priori choice between only sulfate and ash?

The revision quoted in the previous comment should help clarify this. Indeed, for Calbuco we assumed only sulfate or ash was sampled since the plume originates from Chile where the volcano exists and because we are not aware of any stratospheric smoke injections from Australia or South America at this time. We assumed that because these layers had elevated depolarization and originated from the location of the volcano, they are most likely ash and not smoke.

Section 4.1 It's very interesting that the color ratio is not used anymore, and this seems to work well. In the analysis it's suggested that there are fewer misclassifications of known cases without the color ratio than with it. However, in Section 4.1 "Color ratio test for smoke removed", most of the discussion is about why the depolarization ratio is sufficient and there is not much discussion about the color ratio. I hope the authors will consider adding more to illustrate why it is advantageous to take it out. For instance, expected uncertainty levels could be discussed and the distributions of color ratios and percentage of overlap in the color ratio dimension should also be discussed.

Agreed. The crux of the argument for taking out the color ratio test was not well emphasized. The primary rationale is that the secondary mode in the joint distribution for smoke events in V4.2 having relatively low depolarization and high color ratio (Fig. 4) goes away when tropospheric smoke is no longer included. Because this signature is not observed in stratospheric smoke arising from pyroCb events (what we're primarily targeting), then there is no need to intentionally look for these layers. Doing so would cause misclassifications of sulfate as smoke which would cause overestimated in layer AOD when the incorrect lidar ratio is assigned. With this in mind, we made the following adjustments.

The parenthetical reference at the beginning of Section 4.1 instructing the reader to compare Fig. 4 with Fig. 8 is expanded to be more explicit: "The new joint distributions in Fig. 8 show that excluding tropospheric smoke layers from the sample population had an important impact: the population of smoke with low depolarization and higher color ratio is no longer prominent (notice that the secondary mode labeled "tropospheric smoke" in Fig. 4 is not evident in Fig. 8)."

The following sentence is added in the second paragraph of Sect. 4.1: "Because smoke layers having low depolarization/high color ratio are not routinely observed in the stratosphere, then there is no need to search for them."

A discussion about the impacts on AOD accuracy is added to the section: "Removing the color ratio test has the additional effect of allowing more sulfate layers to be classified correctly as sulfate rather than smoke. This improves the accuracy of the retrieved extinction for these sulfate layers because the correct lidar ratio of 50 sr will be selected rather than 70 sr (Sect. 4.5), thereby avoiding a 40 % overestimate in aerosol optical depth (AOD). Conversely, there will be smoke layers misclassified as sulfate due to the overlap in the distributions for these two types, yielding a 40 % underestimate in AOD due to selecting the lower lidar ratio. The relative frequencies of correct and incorrect classification for sulfate and smoke will be evaluated in Sect. 4.6."

Section 4. I believe there's another signature of smoke in CALIPSO data related to the differential attenuation between 532 and 1064 nm (i.e. that the color ratio changes between the top and bottom

of the layer). Is this present for stratospheric as well as tropospheric smoke layers, and is there any thought of using this in the automated classification algorithm(s)?

In the past, the CALIPSO team has sought ways to take advantage of the differential attenuation between 532 nm and 1064 nm to better identify smoke layers. While this signature can be readily identified in some scenarios, particularly when optical depth is high, we found that the CALIOP SNR is too low for many situations to confidently rely on the change in color ratio with layer penetration in automated processing. For example, calculating the slope of color ratio with penetration depth is too variable in low optical depth scenarios to be reliable for all the scenes the CALIPSO encounters. Nonetheless, this signature is present in both stratospheric and tropospheric smoke layers, awaiting use by algorithms for future lidar missions.

L365-366 "The lowest quartile... was selected so that the average relative uncertainty in estimated particulate depolarization is less than 250%" (and following text). While I'm not suprised by very large uncertainties in estimated particulate depolarization ratio for small signal levels, it's not immediately obvious how an uncertainty of more than 100% is particularly useful, and whether it should make any practical difference if the uncertainty is 200% or 400%, especially given that sulfate has essentially zero depolarization and therefore always has a large relative uncertainty. However, the uncertainty in the estimated particulate depolarization is strongly dominated by systematic, rather than random, uncertainty, particularly for small values of the attenuated scattering ratio, due to the mathematical form of Eqn 2 (see, e.g. Burton et al. 2015) which is a strong and valid reason for wanting to cut off this calculation at values before the systematic error blows up. I think the discussion would be clearer and more informative if the discussion explicitly refers to this systematic uncertainty from Eqn 2 and perhaps includes a figure showing how the resulting uncertainty depends on the attenuated backscatter.

Indeed, a visualization would help make the message clearer and add more information. To acknowledge the rapid increase in uncertainty for low IAB values, we added these sentences,

"At low signal levels, $\delta_p^{est}$ becomes prone to increases in systematic and random errors along with the increased risk of overestimates as discussed in Sect. 3….The lowest-quartile $\gamma'_{532}$ metric was selected based on the rapid increase in relative uncertainty in $\delta_p^{est}$ as a function of decreasing $\gamma'_{532}$ (Fig. 12)."

Here is the new Fig. 12 which shows the old and new thresholds and how they relate to the relative uncertainty in $\delta_p^{est}$. Adding this figure adds valuable information, that the average relative uncertainty for stratospheric aerosol layers detected at 5 km resolution is less than 50%. A sentence is added to the section to note this.

[Figure]

**Figure 12.** Average relative uncertainty in estimated particulate depolarization ratios as a function of layer integrated attenuated backscatter ($\gamma'_{532}$) for all stratospheric aerosol layers detected between June 2006 – December 2018, segregated by horizontal averaging required for detection. Solid lines are nighttime detections and dashed lines are daytime detections. Layers classified as PSA, |CAD| < 20, or top altitudes above 20 km are excluded.

On a similar topic, what are the uncertainty and spread in the volume depolarization ratios? Has there been any investigation into whether it's feasible to do stratospheric aerosol classification using the volume depolarization ratio instead of the estimated particulate depolarization ratio? It would be interesting to see a figure like Figure 6 with joint distributions, but with volume depolarization instead of estimated particulate depolarization.

The uncertainty and spread of volume depolarization ratio is smaller than that of the estimated particulate depolarization ratio due to the propagated errors from the attenuated scattering ratios in the latter. The decision to use estimated particulate versus volume depolarization ratios for aerosol subtyping during the formulation of the CALIPSO mission was based on the rationale added to Sect. 3 – to retain the ability to discriminate between depolarizing aerosol (but with low scattering ratios) and non-depolarizing aerosol. Though a substantial amount of analysis went into this decision, including considerations of noise contributions, there is not a single publication where this is documented. Indeed, packaging all this analysis into a dedicated publication would be useful for the lidar community. Because the choice to use particulate instead of volume depolarization is the engrained strategy for all CALIOP aerosol subtyping (tropospheric and stratospheric), we did not investigate the possibility of using volume depolarization ratios for stratospheric aerosol subtyping alone.

L409-410 "because knowledge of 1064 nm lidar ratios is not as broad in the literature". But the new 532 nm lidar ratios were arrived at via constrained retrievals, if I followed that correctly. Isn't that technique just as valid at 1064 nm, regardless of other existing literature?

In theory the technique is valid at 1064 nm, but for the power output of CALIOP there is not sufficient SNR for a reliable retrieval. The backbone of the constrained retrieval is measurements of attenuated scattering ratios (the ratio of particulate + molecular attenuated backscatter to molecular attenuated backscatter) in clear-air above and below the particulate layer. This works very well at shorter wavelengths such as 532 nm because there is a strong backscatter return from the molecular component. However, at 1064 nm, the molecular return is much weaker. This, combined with limitations of the CALIOP power levels, yields scattering ratios with too low of

SNR for reliable constrained retrievals using the 532 nm clear-air technique. The CALIPSO project plans to explore other lidar ratio retrieval techniques for 1064 nm in a future data release.

L430 "increased variability in the estimated particulate depolarization ratios for layers [between] the old and new ... thresholds". It's definitely plausible that there is more uncertainty and more variability in these layers. I would argue further, though, that the magnification of the uncertainty and error in the scattering ratio swamps the magnification of the random error in (volume) depolarization measurements (See Burton et al. 2015 Table 2). And if my speculation earlier (see comment pertaining to line 179-180) has merit, then the systematic error will tend to bias the estimated particulate depolarization to higher values. That might account for some misclassification of sulfate as smoke.

The sentence has been modified to explicitly reference the greater influence of random and systematic errors:

"The increased variability in $\delta_p^{est}$ due to the greater influence of random and systematic errors for layers having $\gamma'_{532}$ in between the old and new low-$\gamma'$ thresholds allows more opportunities for sulfate layers to exceed the $\delta_p^{est} > 0.075$ threshold."

L511 please include a reference and/or more explanation for negative brightness temperature differences being a signature of ash.

A reference to Prata (1989) was added which describes how the absorption for ash layers increases with increasing wavelength in the $10 - 13$ μm range, leading to negative brightness temperature differences.

Prata, A. J.: Infrared radiative transfer calculations for volcanic ash clouds, Geophys. Res. Lett., 16, 1293–1296, 1989

L618-619 "This reflected sunlight enhances noise throughout the profile overhead, thereby increasing the variability of depolarization ratio measurements". It's clear that noise would increase the variability of the depolarization measurements, including volume depolarization, but why does it produce an asymmetric tail with more large values? Is this because the negative branch is cut off (I mean if negative depolarizations were reported and graphed would there be a symmetric tail on that side?) or is it related to the propagation of error from the volume depolarization to the estimated particulate depolarization? By which I mean, if the volume depolarization distributions are plotted, are they wider during the day but still symmetric? If so, then, considering that noise also increases the variability in R', perhaps the issue is again the R' errors propagating non-linearly through Eqn 2, so that a symmetric distribution of error in R' (plus a symmetric distribution of volume depolarization) becomes an asymmetric distribution of particulate depolarization.

The asymmetric tail in the daytime depolarization is inherent to the volume depolarization ratio. The figures below compare the volume and estimated particulate depolarization ratios for the two events in Fig. 21 (previously Fig. 18 in the initial submission). The statistics on the charts show

that the breadth and skewness did not change appreciably or at least enough to confidently attribute the asymmetric tail to amplification of systematic errors in R' through Eqn. (2). However, we agree that the negative depolarization should be shown to clearly represent the distributions. For this, we reduced the lower limit of depolarization on the x-axis of Fig. 21 so that the negative depolarizations are shown.

The cause of the asymmetric tail can be shown by considering the attenuated backscatters comprising the volume depolarization ratio as mean values plus a delta that reflects the fractional contribution of noise to the mean:

$$\delta_v = \frac{\langle\beta'_\perp\rangle(1+\Delta_\perp)}{\langle\beta'_\parallel\rangle(1+\Delta_\parallel)} \tag{R3}$$

Because noise is proportional to the signal strength, and because the perpendicular signal is less than the parallel signal, the denominator in Eqn. R3 is the most influential factor. When the fractional noise in the parallel channel grows to a large positive contribution ($\Delta_\parallel \to 1$), the $\delta_v$ is reduced to about half its true value. When it grows to a large negative contribution ($\Delta_\parallel \to -1$), the $\delta_v$ increases dramatically as the denominator becomes small. This leads preferentially to a positive skew in the depolarization ratio.

[Figure]

L621 I believe this is a bit of a misquote. Burton et al. 2015 included an overview of other lidar observations, some showing low and some showing high depolarization of tropospheric smoke. Some of the high depolarization ones are "aged", but for logistical reasons, lidar observations of fresh smoke is rare, and of aged smoke is relatively common, so both the low and high subsets are dominated by aged smoke. Burton et al. doesn't give any mechanism for aging affecting depolarization, except to quote Martins et al. 2018 who do. The depolarization of smoke in the troposphere might not be so relevant to depolarization of smoke in the stratosphere anyway. However, note that Haarig et al. 2018 (already quoted in this manuscript) also proposes a hypothesis that's related to aging about why stratospheric smoke tends to be depolarizing while tropospheric tends not to be. And for completeness, there's also the ice hypothesis of Kablick et

al. 2018 (Kablick III, et al.: The Great Slave Lake PyroCb of 5 August 2014: Observations, Simulations, Comparisons With Regular Convection, and Impact on UTLS Water Vapor, Journal of Geophysical Research: Atmospheres, 123, 12,332-312,352, 10.1029/2018jd028965, 2018.). Sicard et al. 2019 also have relevant discussion (Sicard, et al.: Ground/space, passive/active remote sensing observations coupled with particle dispersion modelling to understand the inter-continental transport of wildfire smoke plumes, Remote Sens Environ, 232, 111294, https://doi.org/10.1016/j.rse.2019.111294, 2019.)

Agreed. The Burton et al., 2015 reference is not the best choice for attributing the variability of depolarization from stratospheric smoke to ageing. Considering that the last part of this paragraph is all speculative, we chose to remove the discussion about ageing and just state, "The nighttime distributions of $\delta_p^{est}$ in Fig. 18 are expected to more closely resemble natural variability." We also added the references you mentioned to this sentence in section 4.1, which helps to bolster the statement, "The cause of aspherical particles in smoke plumes from these pyroCb events is an active area of research (e.g., Gialitaki et al., 2020; Haarig et al., 2018; Kablick et al., 2018; Sicard et al., 2019)."

L656 "The aerosol subtypes of these false layer detections" I'm confused whether these percentages refer to the whole population of false layer detections, or only to the subgroup that remains after filtering for the SAA and low CAD scores. If they include the low CAD group, then is it likely that the dominance of high-depolarization aerosol types might reflect mistyped cloud? Does the percentage classified as ash remain so remarkably high after the low CAD cases are taken out? If so, why should these non-layers be dominated by large depolarizations?

The percentages of aerosol subtypes of the false layer detections refer to the whole population of false layer detections. To clarify this, the underlined text was added to the sentence below.

"As a first-order estimate, during the year 2013 when there were no major stratospheric aerosol injections, suspected false-positive feature detections (any layer detected above 20 km, excluding PSA) occurred in 0.4 % of profiles at night and 1 % at day, globally."

Yes, these statistics include the low CAD group. It is unlikely that misclassified cloud is the primary cause of high depolarization for these layers. The 20 km altitude restriction is a reasonable limit to avoid high-level clouds, including those that overshoot the tropopause. Suspected false feature detections occur all the way up to the maximum level 2 altitude of 30 km where meteorological clouds surely do not exist. For the 3% of non-PSA features above 20 km having CAD > 20, the percentage classified as ash is somewhat smaller, 57% (day) and < 1% (night). These layers have low confidence CAD scores (66% have |CAD score| < 50, indicating low confidence).

The following figures have been added to the supplement to show representative examples of false feature detections in CALIOP data. The first example Fig. S6 is a false detection due to a radiation-induced current spike in the 532 nm channel detector(s) (Hunt et al., 2009). Because the volume depolarization is so high for this layer, it is likely that the perpendicular channel was affected by the current spike and not the parallel channel. The second example Fig. S7 is daytime over the South Atlantic Anomaly region where excess radiation more commonly affects the CALIOP

detectors. This scene contains false feature detections due to radiation-induced current spikes and from enhanced noise in the column where sunlight is reflecting off cloud tops at lower altitudes. The volume depolarization is noted for each of the false feature detections that are above 20 km and classified as ash. The enhanced depolarization in this case is due to the enhanced noise in the signal which broadens the distributions of perpendicular and parallel backscatter. The ratio of these noisy quantities results in a broad distribution of depolarizations with a tendency toward very high values. The underlined text below was added to Sect. 5.4 as supporting information:

"These layers are primarily located inside the SAA. Outside the SAA, a relatively small number of layers are randomly distributed over the globe (Fig. S5). Because they do not occur within specific latitude bands, as would legitimate layers associated with specific events, they are likely due to false layer detections caused by radiation-induced current spikes in the 532 nm channel detectors (Hunt et al., 2009; e.g., Fig. S6) or enhanced background noise from sunlight reflecting off underlying clouds in the daytime (e.g., Fig. S7). The aerosol subtypes of these false layer detections at night are 54 % ash, 30 % sulfate, 5 % smoke, and 11 % unclassified. During the day, these frequencies change to 84 % ash, <1 % sulfate, 3 % smoke, and 13 % unclassified. The propensity for ash classification is due to excess solar background noise broadening the distribution of depolarization values to create artificially high values and from cases of radiation-induced current spikes which only affect the 532 nm perpendicular channel, but not the parallel channel."

[Figure]

**Figure S6.** Nighttime example of (a) CALIOP 532 nm attenuated backscatter and (b) V4.5 feature detection & classification for the granule 2013-01-01T16-52-04ZN. A radiation-induced current spike is noted at 24 km which causes a false feature detection in the stratosphere. The volume depolarization for the layer is noted in panel (b).

[Figure]

**Figure S7.** Daytime example over the South Atlantic Anomaly region of (a) CALIOP 532 nm attenuated backscatter and (b) V4.5 feature detection & classification for the granule 2013-02-11T17-35-36ZD. False feature detections occur in stratosphere due to radiation-induced current spikes and increased noise in the column due to sunlight reflecting off underlying clouds. The volume depolarization for a several false feature detections are noted in panel (b).

Comments on presentation details

L23-24 "more likely". More likely than what? More likely than the previous algorithm, or more likely to be correctly classified than misclassified?

The sentence is changed to "…it is more likely that layers labeled as this subtype are in fact sulfate compared to those given the sulfate/other classification in the previous data release."

L112 why "primary"? Are there more resolutions than these three?

Since this paragraph is describing CALIOP level 2 processing in general, the sentence is changed to "The horizontal averaging resolutions for layer detection are 1/3 km, 1 km, 5 km, 20 km, and 80 km." The paper focuses on the 5, 20, and 80 km resolutions because these are the resolutions where extinction profiles are reported and because 1/3 km resolution features are only detected below 8.2 km (the downlinked resolution is coarser above that altitude). The 1 km resolution is an intermediate step to find 1/3 km layer detections; the Vaughan et al., 2009 reference in the sentence goes into these details. In terms of stratospheric aerosol detection, the 5, 20, and 80 km resolutions are the most relevant.

L145 I don't know what the "1064 nm baseline shape" means.  I understand it is not important to the topic to your paper but since it's mentioned, it would be good to have enough information  to understand what it means.

A short definition of the baseline shape has been added to the sentence and the section is modified slightly to alert the reader that details of this information can be found in the CALIPSO Lidar Level 1 V4.51 Data Quality Statement (2022) reference earlier in the paragraph.

"The V4.51 level 1B data used as input for the V4.5 level 2 products also contains several updates relative to the previous release, described in the CALIPSO Lidar Level 1 V4.51 Data Quality Statement (2022): The 532 nm daytime and 1064 nm calibration algorithms now mitigate the influence of low energy laser shots on the derived calibration coefficients. This corrects biases and reduces calibration uncertainty in these channels at SAA latitudes (~15° S to 30° S) since the onset of low energy shots in mid-2016. Small corrections have also been made to the 1064 nm baseline shape (the shape of the profile measured by the detectors when the laser is not firing) having negligible impacts on the current analysis."

L149-152 Here in the discussion about counting layers, I wondered about whether changing layer boundaries between versions would impact the analysis. Later I can see that it really doesn't, but an explanation in this paragraph about why lack of perfect layer matching doesn't impact the analysis would be helpful.

Adding this information is certainly helpful. However it would be best to add this information to later sections rather than this particular paragraph. To understand why perfect layer matching isn't necessary for our analysis method requires a description of the plume boundaries that we use to identify layers. That appears in the first paragraph of Sect. 3 and the comparison of V4.2 to V4.5 is in Sect. 4.6. To help clarify what we've done and to link these points together, the following revisions are made.

In Sect. 3, the description of plume boundaries is better explained. The new text is underlined below.

"Plumes are tracked manually in CALIOP imagery over successive days and their latitude/longitude/altitude boundaries are recorded for each CALIOP granule. The "plume boundaries" we select are rectangles of altitude × along-track distance that encompass the plume, plus a ~1 km buffer of "clear-air" where no other features are detected. In order to avoid cloud contamination, plumes near or in contact with high altitude cirrus or overshooting cloud tops are excluded. All layers detected in the level 2 aerosol layer product within the rectangular plume boundaries contribute to the joint distributions. The full list of CALIOP granules and plume boundaries for all events analyzed for V4.2 and V4.5 development is reported in the Supplement"

In Sect. 4.6, the text references back to the plume boundary description. New text is underlined.

"The changes in classifications due to V4.5 revisions are summarized in Fig. 14 based on all the manually classified events in Table 1. Here we are comparing unique layers detected within the plume boundaries reported in the Supplement. Because each rectangular boundary contains a

buffer of clear-air around the actual plume (Sect. 3), it is not necessary to perfectly match layer top and base altitudes between versions. We are only concerned with the classifications within those boundaries, and the buffer allows for any differences in layer detection that may occur."

L274 "their composition was determined". Add references.

These references were added, along with revisions to the text that are described in the Specific comments on content above related to L274: Prata et al., 2014; Andersson et al., 2013, Marzano et al., 2018, Bègue et al., 2017.

Figure 5 flowchart. In the decision about the attenuated backscatter on the top row, third diamond from the left, I believe < should be > here.

Yes! You are absolutely right. That would have been an unfortunate typo. It is now corrected. Thank you!

L362 "confidently classified" is confusing since the reason for the cutoff is that they can't be confidently classified. Did "confidently classified" refer to the cloud-aerosol discrimination? Can this be made clearer?

Agreed, the phrase "confidently classified" is not clear. The phrase tied back to the last sentence of section 4.3 which asserts that layers with insufficient backscatter cannot be confidently classified as ash, smoke, or sulfate based on depolarization. It has been removed from the sentence, which is now: "Previously the threshold was $\gamma'_{532} < 0.001$ sr$^{-1}$ which caused 75 % of all stratospheric aerosol layers detected from June 2006–December 2018 to be classified as sulfate/other."

L368 "those layers" I'm having trouble following. Which layers? Does this refer to daytime/80km layers that have attenuated backscatter between the old and new cutoffs? If they can't be confidently classified, should they be left in the unclassified category?

The sentence is clarified as, "Classifications for daytime, 80 km resolution stratospheric aerosol layers that are not assigned the unclassified subtype should be interpreted with caution."

L369 "these classifications". Again, which ones? The 80 km layers? Or all the layers between the old and new thresholds?

The sentence is clarified as, "For reference, the average relative uncertainty in $\delta_p^{est}$ was less than 200 % for stratospheric aerosol layers having $\gamma'_{532}$ above the previous low-$\gamma'$ threshold in V4.2."

L372 "increase of 50% in the relative uncertainty". This is ambiguous. It could mean a relative increase of 50% or an absolute increase of 50 percentage points. I think the latter is what's intended. If so, consider "50 percentage points".

The latter was intended. The text is changed to "50 percentage points".

Figure 15d. Why is the time series cut off while the sulfate signature is still high?

This was a limitation on the V4.5 test data that we generated for the CALIPSO project's final quality assessment prior to release. We only generated around three years of data in total so that limited what is shown – in the case of Nabro the project happened to generate more than just two weeks after the eruption. At a minimum for this paper, we ensured that the analysis covers at least the first two weeks since the first CALIOP detection.

Figure 16. It's somewhat confusing that this shows volume depolarization ratio when the text primarily discusses estimated particulate depolarization ratio.

We prefer to show the volume depolarization ratio because it shows the range-resolved structure of depolarization. The estimated particulate depolarization ratio is only computed based on layer-averaged quantities in CALIOP processing, so the variability in depolarization would not be as obvious if estimated particulate depolarization ratio were plotted. In order to better give the sense of its value, call-outs were added to each figure giving representative values of estimated particulate depolarization for different parts of each plume.

[Figure]

L673. "threshold ... to discriminate between volcanic ash and sulfate". I think this is a typo, as these two types do not share a boundary.

A typo indeed! Thanks for catching that…it is now "to discriminate between volcanic ash and smoke".

L693. "smoke transported". Consider "any smoke transported" or "if smoke is transported"

Changed to "any smoke transported".

L708. Add reference for the split window technique.

We chose to eliminate the sentence that mentioned the split window technique. At this point in the conclusion, we're discussing how passive instrument measurements in combination with lidar observations might be able to help discriminate among ash, sulfate, and smoke in the troposphere without getting too specific. The split-window technique is useful for discriminating between volcanic ash and ice clouds (not between smoke and ash as stated in the sentence previously). Since this is a bit off topic of the point we're trying to make, we found it is best to remove the sentence entirely.

**Note from the authors on additional changes to the manuscript**

- The release date for the V4.5 level 2 data products has changed from 2022 to early 2023. This is now reflected in Sect. 2 and in the Data Availability section.

- We changed the notation for attenuated scattering ratio, discussed in Sect. 2, from $R_{mas}$ to $R'$. The previous notation was used for consistency with that of Omar et al., 2009. However, we find the $R'$ notation is clearer because the prime indicates it is an attenuated quantity, the consequences of which are now discussed in the section. Further, $R'$ is a common notation for attenuated scattering ratio within the lidar community and is used throughout CALIOP algorithm theoretical basis documents.

---

## Author Comment (AC2)

**Response to Anonymous Referee #2 Comments**

Authors provide description of updated version of stratospheric aerosol subtyping algorithm (version 4.5) for CALIOP. They provide very detailed explanation of the reasons to revise the lidar ratios and the lidar ratios, as well as changes in algorithm structure. Updated algorithn is applied to numerous measurement cases, corresponding to ash, smoke and sulfate dominance and results are compared with previous version (V 4.2), demonstrating the difference. The manuscript is well and clearly written, thus is suitable for publishing in AMT. The results presented will be useful for scientific community studying the stratospheric aerosol.

Referee #1 provided detailed comments on manuscript and I have not much to add. But I am confused with choice of lidar ratio for smoke at 1064 nm (Table 2). The value of 30 sr that they suggest is very low by my opinion. There are several publication of Leipzig group

Depolarization and lidar ratios at 355, 532, and 1064 nm and microphysical properties of aged tropospheric and stratospheric Canadian wildfire smoke, Moritz Haarig, Albert Ansmann, Holger Baars, Cristofer Jimenez, Igor Veselovskii, Ronny Engelmann, and Dietrich Althausen, Atmos. Chem. Phys., 18, 11847–11861, https://doi.org/10.5194/acp-18-11847-2018, 2018

The lidar ratios there are 40–45 sr (355 nm), 65–80 sr (532 nm), and 80–95 sr (1064 nm) for Canadian smoke….

Australian wildfire smoke in the stratosphere: the decay phase in 2020/2021 and impact on ozone depletion, Kevin Ohneiser, Albert Ansmann, Bernd Kaifler, Alexandra Chudnovsky, Boris Barja, Daniel A. Knopf, Natalie Kaifler, Holger Baars, Patric Seifert, Diego Villanueva, Cristofer Jimenez, Martin Radenz, Ronny Engelmann, Igor Veselovskii, and Félix Zamorano, Atmos. Chem. Phys., 22, 7417–7442, https://doi.org/10.5194/acp-22-7417-2022, 2022

Combined lidar–photometer retrievals revealed typical smoke extinction-to-backscatter ratios of $69 \pm 19$ sr (at 355 nm), $91 \pm 17$ sr (at 532 nm), and $120 \pm 22$ sr (at 1064 nm) for Australian smoke.

I think this difference should be commented.

Thank you for your review and for your suggestion about the choice of lidar ratio for smoke at 1064 nm. We added the following paragraph to Sect. 4.5 to address these observations.

"Recent ground-based lidar and lidar-photometer retrievals of smoke arising from pyroCb events have measured higher 1064 nm lidar ratios than the default value used by CALIOP (30 sr), with values ranging from 80 – 120 sr (Haarig et al., 2018; Ohneiser et al., 2022). The 1064 nm lidar ratio used for all smoke layers in V4.2, and carried forward into V4.5, is based on AERONET retrievals of tropospheric smoke (Sayer et al., 2014). Microphysical differences likely exist between smoke injected into the stratosphere from pyroCb events and smoke residing in the troposphere from more docile events, so lidar ratio differences are plausible. We plan to reevaluate 1064 nm lidar ratios for stratospheric smoke and ash in a future data release."

**Note from the authors on additional changes to the manuscript**

- The release date for the V4.5 level 2 data products has changed from 2022 to early 2023. This is now reflected in Sect. 2 and in the Data Availability section.

- We changed the notation for attenuated scattering ratio, discussed in Sect. 2, from $R_{mas}$ to $R'$. The previous notation was used for consistency with that of Omar et al., 2009. However, we find the $R'$ notation is clearer because the prime indicates it is an attenuated quantity, the consequences of which are now discussed in the section. Further, $R'$ is a common notation for attenuated scattering ratio within the lidar community and is used throughout CALIOP algorithm theoretical basis documents.

---

## Author Comment (AC3)

**Response to Anonymous Referee #3 Comments**

The manuscript "The CALIPSO version 4.5 stratospheric aerosol subtyping algorithm" present the changes in the stratospheric aerosol subtyping algorithm, as well as the performance assessment of the updated algorithm for several events. The use of integrated attenuated backscatter color ratio was eliminated, the values of depolarization threshold and low-r' threshold were revised, and ash lidar ratio was updated. The subtypes are also revised, "sulfate" and "unclassified" are used instead of "sulfate/other". Results are promising, and the stratospheric aerosol subtyping is improved. The dataset is interesting, and the manuscript is well written. The manuscript is worthwhile to be published, after addressing all the points raised by reviewers.

In the current version (v4.5), only products from 532 nm are used for the typing (for ash, smoke, sulfate and unclassified), and 1064 nm is not used as it does not seem to provide additional improvement in aerosol typing. Results show removing the color ratio would improve the accuracy of typing. The current aerosol typing is similar by using a single wavelength polarization elastic lidar. Maybe authors can comment on the advantage of using multiple wavelengths, which could be useful for future satellite missions.

Thank you for your excellent suggestions and corrections that you noted. For the request above, indeed, using multiple wavelengths could be useful for stratospheric aerosol subtyping with future missions. The concluding paragraph touches on this: "More sophisticated instrumentation will improve discrimination capability in the troposphere, such as high spectral resolution lidar with depolarization sensitivity at 355 nm, 532 nm, and 1064 nm (e.g., as in Burton et al., 2015) or combined lidar plus passive instrument retrievals." The requested changes for minor comments below have all been implemented.

Please see below some minor comments:

"particulate depolarization ratio" is used in the figures, but the parameter used in the study is the "estimated particulate depolarization ratio", maybe change them to "estimated particulate depolarization ratio" for the clarity (fig3, 6, 7, 8, 13, 18)

The text in the figures is modified to estimated particulate depolarization ratio as suggested.

P7 l181, add "at 532 nm" for the molecular depol

Added. Good call!

P13 l307-308, High depolarization ratio suggests the aspherical shape. Can you clarify which observations suggest the smaller size of smoke particles at higher altitudes, color ratio?

The sentence was clarified as follows, "These observations, primarily representing pyroCb events, suggest that smoke injected to extremely high altitudes contains particles that are aspherical and smaller (based on enhanced depolarization and lower color ratios, respectively) compared to smoke injected to lower altitudes."

P17 l407, why using 20 as the threshold of |CAD score|?

A |CAD score| of less than 20 indicates that the cloud-aerosol discrimination has no confidence in the determination of if the layer is a cloud or an aerosol. It is a standard quality assurance practice to exclude layers having these CAD scores to avoid introducing cloud contamination. In order to make our intention clearer, the sentence is modified as, "In order to remove no-confidence retrievals and any possible cloud contamination, only layers with retrieved lidar ratio uncertainty < 100 % and |CAD score| > 20 contribute to the histogram."

P19 l456, "." is missing before "There is a small".

Added

P19 fig11, the low-r' layers were defined using different thresholds for v4.2 and v4.5, it would good to add such information in the caption.

The caption has been revised to "Low-$\gamma'$ layers, based on the $\gamma'$ threshold for the relevant version, are indicated by hatched bars."

P20 l471, in the Fig.S1,3,4, which parameter of unique layers was used, layer mid/top or top-to-base?

The layer top and layer base were used to determine the altitudes that each unique layer spanned. To make this clearer, the captions for Figs. S1, S3, and S4 are changed to: "Horizontal axis indicates number of aerosol samples within each 100 m range bin based on the top and base altitude of each unique layer."

P21 fig12, clarify the version in the caption. (a) change 30 N to 30 S

The caption was changed to "For ash-dominated events in V4.5…" and the latitude limits are corrected (thanks for catching that).

P24 fig15, (d) y-label is missing

The y-label has been added.

P27 l617, change PBL to planetary boundary layer (only used once)

Changed to planetary boundary layer.

**Note from the authors on additional changes to the manuscript**
- The release date for the V4.5 level 2 data products has changed from 2022 to early 2023. This is now reflected in Sect. 2 and in the Data Availability section.

- We changed the notation for attenuated scattering ratio, discussed in Sect. 2, from $R_{mas}$ to $R'$. The previous notation was used for consistency with that of Omar et al., 2009. However, we find the $R'$ notation is clearer because the prime indicates it is an attenuated quantity, the consequences of which are now discussed in the section. Further, $R'$ is a common notation for attenuated scattering ratio within the lidar community and is used throughout CALIOP algorithm theoretical basis documents.